# Stochastic Multiple Target Sampling Gradient Descent

**Hoang Phan**[1]  **Ngoc N. Tran**[1]  **Trung Le**[2]  **Toan Tran**[1]  **Nhat Ho**[3]  **Dinh Phung**[1,2]

[1] VinAI Research, Vietnam
[2] Monash University, Australia
[3] University of Texas, Austin

## Abstract

Sampling from an unnormalized target distribution is an essential problem with many applications in probabilistic inference. Stein Variational Gradient Descent (SVGD) has been shown to be a powerful method that iteratively updates a set of particles to approximate the distribution of interest. Furthermore, when analysing its asymptotic properties, SVGD reduces exactly to a single-objective optimization problem and can be viewed as a probabilistic version of this single-objective optimization problem. A natural question then arises: "*Can we derive a probabilistic version of the multi-objective optimization?*". To answer this question, we propose *Stochastic Multiple Target Sampling Gradient Descent* (MT-SGD), enabling us to sample from multiple unnormalized target distributions. Specifically, our MT-SGD conducts a flow of intermediate distributions gradually orienting to multiple target distributions, which allows the sampled particles to move to the joint high-likelihood region of the target distributions. Interestingly, the asymptotic analysis shows that our approach reduces exactly to the multiple-gradient descent algorithm for multi-objective optimization, as expected. Finally, we conduct comprehensive experiments to demonstrate the merit of our approach to multi-task learning.

## 1 Introduction

Sampling from an unnormalized target distribution that we know the density function up to a scaling factor is a pivotal problem with many applications in probabilistic inference [2, 18, 25]. For this purpose, Markov chain Monte Carlo (MCMC) has been widely used to draw approximate posterior samples, but unfortunately, is often time-consuming and has difficulty accessing the convergence [14]. Targeting an efficient acceleration of MCMC, some stochastic variational particle-based approaches have been proposed, notably Stochastic Langevin Gradient Descent [26] and Stein Variational Gradient Descent (SVGD) [14]. Outstanding among them is SVGD, with a solid theoretical guarantee of the convergence of the set of particles to the target distribution by maintaining a flow of distributions. More specifically, SVGD starts from an arbitrary and easy-to-sample initial distribution and learns the subsequent distribution in the flow by push-forwarding the current one using a function $T(x) = x + \epsilon\phi(x)$, where $x \in \mathbb{R}^d$, $\epsilon > 0$ is the learning rate, and $\phi \in \mathcal{H}_k^d$ with $\mathcal{H}_k$ to be the Reproducing Kernel Hilbert Space corresponding to a kernel $k$. It is well-known that for the case of using Gaussian RBF kernel, by letting the kernel width approach $+\infty$, the update formula of SVGD at each step asymptotically reduces to the typical gradient descent (GD) [14], showing the connection between a probabilistic framework like SVGD and a single-objective optimization algorithm. In other words, SVGD can be viewed as a probabilistic version of the GD for single-objective optimization.

On the other side, multi-objective optimization (MOO) [7] aims to optimize a set of objective functions and manifests itself in many real-world applications problems, such as in multi-task learning (MTL) [17, 24], natural language processing [1], and reinforcement learning [9, 22, 21]. Leveraging the above insights, it is natural to ask: "*Can we derive a probabilistic version of multi-*

*objective optimization?*". By answering this question, we enable the application of the Bayesian inference framework to the tasks inherently fulfilled by the MOO framework.

**Contribution.** In this paper, we provide an affirmative answer to that question. In particular, we go beyond the SVGD to propose *Stochastic Multiple Target Sampling Gradient Descent* (MT-SGD), enabling us to sample from multiple target distributions. By considering the push-forward map $T(x) = x + \epsilon\phi(x)$ with $\phi \in \mathcal{H}_k^d$, we can find a closed-form for the optimal push-forward map $T^*$ pushing the current distribution on the flow simultaneously closer to all target distributions. Similar to SVGD, in the case of using Gaussian RBF kernel, when the kernel width approaches $+\infty$, MT-SGD reduces exactly to the multiple-gradient descent algorithm (MGDA) [7] for multi-objective optimization (MOO). Our MT-SGD, therefore, can be considered as a probabilistic version of the GD multi-objective optimization [7] as expected.

Additionally, in practice, we consider a flow of empirical distributions, in which, each distribution is presented as a set of particles. Our observations indicate that MT-SGD globally drives the particles to close to all target distributions, leading them to diversify on the *joint high-likelihood region* for all distributions. It is worth noting that, different from other multi-particle approaches [13, 15, 17] leading the particles to diversify on a Pareto front, our MT-SGD orients the particle to diversify on the so-called *Pareto common* (i.e., the joint high-likelihood region for all distributions) (cf. Section 2.4 for more discussions). We argue and empirically demonstrate that this characteristic is essential for the Bayesian setting, whose main goal is to estimate the *ensemble accuracy* and the *uncertainty calibration* of a model. In summary, we make the following contributions in this work:

- Propose a principled framework that incorporates the power of Stein Variational Gradient Descent into multi-objective optimization. Concretely, our method is motivated by the theoretical analysis of SVGD, and we further derive the formulation that extends the original work and allows to sample from multiple unnormalize distributions.

- Demonstrate our algorithm is readily applicable in the context of multi-task learning. The benefits of MT-SGD are twofold: i) the trained network is optimal, which could not be improved in any task without diminishing another, and ii) there is no need for predefined preference vectors as in previous works [13, 17], MT-SGD implicitly learns diverse models universally optimizing for all tasks.

- Conduct comprehensive experiments to verify the behaviors of MT-SGD and demonstrate the superiority of MT-SGD to the baselines in a Bayesian setting, with higher ensemble performances and significantly lower calibration errors.

**Related works.** The work of [7] proposed a multi-gradient descent algorithm for multi-objective optimization (MOO) which opens the door for the applications of MOO in machine learning and deep learning. Inspired by [7], MOO has been applied in multi-task learning (MTL) [17, 24], few-shot learning [4, 28], and knowledge distillation [5, 8]. Specifically, in an earlier attempt at solving MTL, [24] viewed multi-task learning as a multi-objective optimization problem, where a task network consists of a shared feature extractor and a task-specific predictor. In another study, [17] developed a gradient-based multi-objective MTL algorithm to find a set of solutions that satisfies the user preferences. Also follows the idea of learning neural networks conditioned on pre-defined preference vectors, [13] proposed Pareto MTL, aiming to find a set of well-distributed Pareto solutions, which can represent different trade-offs among different tasks. Recently, the work of [15] leveraged MOO with SVGD [14] and Langevin dynamics [26] to diversify the solutions of MOO. In another line of work, [28] proposed a bi-level MOO that can be applied to few-shot learning. Furthermore, a somewhat different result was proposed, [8] applied MOO to enable the knowledge distillation from multiple teachers and find a better optimization direction in training the student network.

**Outline.** The paper is organized as follows. In Section 2, we first present our theoretical contribution by reviewing the formalism and providing the point of view adopted to generalize SVGD in the context of MOO. Then, Section 3 introduces an algorithm to showcase the application of our proposed method in the multi-task learning scenario. We report the results of extensive experimental studies performed on various datasets that demonstrate the behaviors and efficiency of MT-SGD in Section 4. Finally, we conclude the paper in Section 5. The complete proofs and experiment setups are deferred to the supplementary material.

## 2 Multi-Target Sampling Gradient Descent

We first briefly introduce the formulation of the multi-target sampling in Section 2.1. Second, Section 2.2 presents our theoretical development and shows how our proposed method is applicable to this problem. Finally, we detail how to train the proposed method in Section 2.3 and highlight key differences between our method and related work in Section 2.4.

### 2.1 Problem Setting

Given a set of target distributions $p_{1:K}(\theta) := \{p_1(\theta), \dots, p_K(\theta)\}$ with parameter $\theta \in \mathbb{R}^d$, we aim to find the optimal distribution $q^* \in \mathcal{Q}$ that minimizes a vector-valued objective function whose $k$-th component is $D_{KL}(q\|p_k)$:

$$\min_{q \in \mathcal{Q}} \left[ D_{KL}(q\|p_1), \dots, D_{KL}(q\|p_K) \right], \tag{1}$$

where $D_{KL}$ represents Kullback-Leibler divergence and $\mathcal{Q}$ is a family of distributions.

When there exists an objective function that conflicts with each other, there will be a trade-off between these two objectives. Therefore, no "optimal solution" exists in such cases. Alternatively, we are often interested in seeking a set of solutions such that each does not have any better solution (i.e. achieves lower loss values in all objectives) [13, 15, 24]. The optimization problem (OP) in (1) thus can be viewed as a multi-objective OP [7] on the probability distribution space. Let us denote $\mathcal{H}_k$ by the Reproducing Kernel Hilbert Space (RKHS) associated with a positive semi-definite (p.s.d.) kernel $k$, and $\mathcal{H}_k^d$ by the $d$-dimensional vector function:

$$f = [f_1, \dots, f_d], (f_i \in \mathcal{H}_k).$$

Inspired by [14], we construct a flow of distributions $q_0, q_1, \dots, q_L$ departed from a simple distribution $q_0$, that gradually move closer to all the target distributions. In particular, at each step, assume that $q$ is the current obtained distribution, and the goal is to learn a transformation $T = id + \epsilon \phi$ so that the *feed-forward distribution* $q^{[T]} = T \# q$ moves closer to $p_{1:K}$ simultaneously. Here we use $id$ to denote the identity operator, $\epsilon > 0$ is a step size, and $\phi \in \mathcal{H}_k^d$ is a velocity field. Particularly, the problem of finding the optimal transformation $T$ for the current step is formulated as:

$$\min_\phi \left[ D_{KL}\left(q^{[T]}\|p_1\right), \dots, D_{KL}\left(q^{[T]}\|p_K\right) \right]. \tag{2}$$

### 2.2 Our Theoretical Development

It is worth noting that the transformation $T$ defined above is injective when $\epsilon$ is sufficiently small [14]. We consider each $D_{KL}\left(q^{[T]}\|p_i\right), i = 1, \dots, K$ as a function w.r.t. $\epsilon$, by applying the first-order Taylor expansion at 0, we have:

$$D_{KL}\left(q^{[T]}\|p_i\right) = D_{KL}(q\|p_i) + \nabla_\epsilon D_{KL}\left(q^{[T]}\|p_i\right)\Big|_{\epsilon=0} \epsilon + O\left(\epsilon^2\right),$$

where $\lim_{\epsilon \to 0} O\left(\epsilon^2\right)/\epsilon^2 = const$.

Given that the velocity field $\phi \in \mathcal{H}_k^d$, similar to [14], the gradient $\nabla_\epsilon D_{KL}\left(q^{[T]}\|p_i\right)\big|_{\epsilon=0}$ can be calculated as provided in [1]

$$\nabla_\epsilon D_{KL}\left(q^{[T]}\|p_i\right)\Big|_{\epsilon=0} = - \langle \phi, \psi_i \rangle_{\mathcal{H}_k^d},$$

where $\psi_i(\cdot) = \mathbb{E}_{\theta \sim q}\left[k(\theta, \cdot)\nabla_\theta \log p_i(\theta) + \nabla_\theta k(\theta, \cdot)\right]$ and $\langle \cdot, \cdot \rangle_{\mathcal{H}_k^d}$ is the dot product in the RKHS.

This means that, for each target distribution $p_i$, the steepest descent direction is $\phi_i^* = \psi_i$, in which the KL divergence of interest $D_{KL}\left(q^{[T]}\|p_i\right)$ gets decreased roughly by $-\epsilon \|\phi_i^*\|_{\mathcal{H}_k^d}^2$ toward the target distribution $p_i$. However, this only guarantees a divergence reduction for a single target distribution $p_i$ itself. Our next aim is hence to find a common direction $\phi^*$ to reduce the KL divergences w.r.t. all target distributions, which is reflected in the following lemma, showing us how to combine the individual steepest descent direction $\phi_i^* = \psi_i$ to yield the optimal direction $\phi^*$ as summarized in Figure 1.

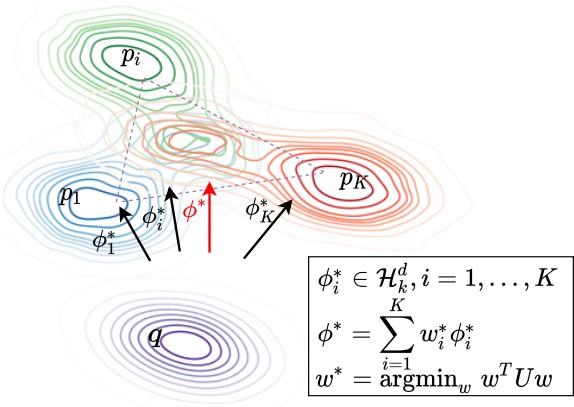

$$\phi_i^* \in \mathcal{H}_k^d, i = 1, \ldots, K$$
$$\phi^* = \sum_{i=1}^{K} w_i^* \phi_i^*$$
$$w^* = \operatorname{argmin}_w w^T U w$$

Figure 1: How to find the optimal descent direction $\phi^*$.

**Lemma 1.** *Let $w^*$ be the optimal solution of the optimization problem $w^* = \operatorname*{argmin}_{w \in \Delta_K} w^T U w$ and $\phi^* = \sum_{i=1}^{K} w_i^* \phi_i^*$, where $\Delta_K = \left\{ \pi \in \mathbb{R}_+^K : \|\pi\|_1 = 1 \right\}$ and $U \in \mathbb{R}^{K \times K}$ with $U_{ij} = \left\langle \phi_i^*, \phi_j^* \right\rangle_{\mathcal{H}_k^d}$, then we have*

$$\left\langle \phi^*, \phi_i^* \right\rangle_{\mathcal{H}_k^d} \geq \|\phi^*\|_{\mathcal{H}_k^d}^2, i = 1, \ldots, K.$$

Lemma 1 provides a common descent direction $\phi^*$ so that all KL divergences w.r.t. the target distributions are consistently reduced by $\epsilon \|\phi^*\|_{\mathcal{H}_k^d}^2$ roughly and Theorem 2 confirms this argument.

**Theorem 2.** *If there does not exist $w \in \Delta_K$ such that $\sum_{i=1}^{K} w_i \phi_i^* = 0$, given a sufficiently small step size $\epsilon$, all KL divergences w.r.t. the target distributions are strictly decreased by at least $A \|\phi^*\|_{\mathcal{H}_k^d}^2 > 0$ where $A$ is a positive constant.*

The next arising question is how to evaluate the matrix $U \in \mathbb{R}^{K \times K}$ with $U_{ij} = \left\langle \phi_i^*, \phi_j^* \right\rangle_{\mathcal{H}_k^d}$ for solving the quadratic problem: $\min_{w \in \Delta_K} w^T U w$. To this end, using some well-known equalities in the RKHS[1], we arrive at the following formula:

$$U_{ij} = \left\langle \phi_i^*, \phi_j^* \right\rangle_{\mathcal{H}_k^d} = \mathbb{E}_{\theta, \theta' \sim q} \Bigg[ k(\theta, \theta') \left\langle \nabla \log p_i(\theta), \nabla \log p_j(\theta') \right\rangle$$
$$+ \left\langle \nabla \log p_i(\theta), \frac{\partial k(\theta, \theta')}{\partial \theta'} \right\rangle + \left\langle \nabla \log p_j(\theta'), \frac{\partial k(\theta, \theta')}{\partial \theta} \right\rangle + \operatorname{tr}\left( \frac{\partial^2 k(\theta, \theta')}{\partial \theta \partial \theta'} \right) \Bigg], \quad (3)$$

where $\operatorname{tr}(\cdot)$ denotes the trace of a (square) matrix.

### 2.3 Algorithm for MT-SGD

For the implementation of MT-SGD, we consider $q$ as a discrete distribution over a set of $M$, ($M \in \mathbb{N}^*$) particles $\theta_1, \theta_2, \ldots, \theta_M \sim q$. The formulation to evaluate $U_{ij}$ in Equation. (3) becomes:

$$U_{ij} = \frac{1}{M^2} \sum_{a=1}^{M} \sum_{b=1}^{M} \Bigg[ k(\theta_a, \theta_b) \left\langle \nabla \log p_i(\theta_a), \nabla \log p_j(\theta_b) \right\rangle + \left\langle \nabla \log p_i(\theta_a), \frac{\partial k(\theta_a, \theta_b)}{\partial \theta_b} \right\rangle$$
$$+ \left\langle \nabla \log p_j(\theta_b), \frac{\partial k(\theta_a, \theta_b)}{\partial \theta_a} \right\rangle + \operatorname{tr}\left( \frac{\partial^2 k(\theta_a, \theta_b)}{\partial \theta_a \partial \theta_b} \right) \Bigg]. \quad (4)$$

The optimal solution $\phi_i^*$ then can be computed as:

$$\phi_i^*(\cdot) = \frac{1}{M} \sum_{j=1}^{M} \left[ k(\theta_j, \cdot) \nabla_{\theta_j} \log p_i(\theta_j) + \nabla_{\theta_j} k(\theta_j, \cdot) \right]. \quad (5)$$

---

[1] All proofs and derivations can be found in the supplementary material.

The key steps of our MT-SGD are summarized in Algorithm 1, where the set of particles $\theta_{1:M}$ is updated gradually to approach the multiple distributions $p_{1:K}$. Furthermore, the update formula consists of two terms: (i) the first term (i.e., relevant to $k\left(\theta_j, \cdot\right) \nabla_{\theta_j} \log p_i\left(\theta_j\right)$) helps to push the particles to the *joint high-likelihood region* for all distributions and (ii) the second term (i.e., relevant to $\nabla_{\theta_j} k\left(\theta_j, \cdot\right)$) which is a *repulsive term* to push away the particles when they reach out each other. Finally, we note that our proposed MT-SGD can be applied in the context where we know the target distributions up to a scaling factor (e.g., in the posterior inference).

---

**Algorithm 1** Pseudocode for MT-SGD.

---

**Input:** Multiple unnormalized target densities $p_{1:K}$.
**Output:** The optimal particles $\theta_1, \theta_2, \ldots, \theta_M$.
1: Initialize a set of particles $\theta_1, \theta_2, \ldots, \theta_M \sim q_0$ .
2: **for** $t = 1$ to $L$ **do**
3:     Form the matrix $U \in \mathbb{R}^{K \times K}$ with the element $U_{ij}$ computed as in Equation. (4).
4:     Solve the QP $\min_{w \in \Delta_K} w^T U w$ to find the optimal weights $w^* \in \Delta_K$.
5:     Compute the optimal direction $\phi^*\left(\cdot\right) = \sum_{i=1}^{K} w_i^* \phi_i^*\left(\cdot\right)$, where $\phi_i^*$ is defined in Equation. (5).
6:     Update $\theta_i = \theta_i + \epsilon \phi^*\left(\theta_i\right), i = 1, ..., K$.
7: **end for**
8: **return** $\theta_1, \theta_2, \ldots, \theta_M$.

---

**Analysis for the case of RBF kernel.** We now consider a radial basis-function (RBF) kernel of bandwidth $\sigma$: $k\left(\theta, \theta'\right) = \exp\left\{-\left\|\theta - \theta'\right\|^2 / \left(2\sigma^2\right)\right\}$ and examine some asymptotic behaviors.

▶ **General case:** The elements of the matrix $U$ become

$$U_{ij} = \mathbb{E}_{\theta, \theta' \sim q}\left[\exp\left\{\frac{-\left\|\theta - \theta'\right\|^2}{2\sigma^2}\right\}\left[\left\langle \nabla \log p_i(\theta), \nabla \log p_j(\theta')\right\rangle\right.\right.$$
$$\left.\left. + \frac{1}{\sigma^2}\left\langle \nabla \log p_i(\theta) - \nabla \log p_j(\theta'), \theta - \theta'\right\rangle - \frac{d}{\sigma^2} - \frac{\left\|\theta - \theta'\right\|^2}{\sigma^4}\right]\right].$$

▶ **Single particle distribution** $q = \delta_\theta$**:** The elements of the matrix $U$ become
$$U_{ij} = \left\langle \nabla \log p_i(\theta), \nabla \log p_j(\theta)\right\rangle,$$
and our formulation reduces exactly to MOO in [7].

▶ **When** $\sigma \to \infty$**:** The elements of the matrix $U$ become

$$U_{ij} = \mathbb{E}_{\theta, \theta' \sim q}\left[\left\langle \nabla \log p_i(\theta), \nabla \log p_j(\theta')\right\rangle\right].$$

## 2.4 Comparison to MOO-SVGD and Other Works

The most closely related work to ours is MOO-SVGD [15]. In a nutshell, ours is principally different from that work and we show a fundamental difference between our MT-SGD and MOO-SVGD in Figure 2. Our MT-SGD navigates the particles from one distribution to another distribution consecutively with a theoretical guarantee of globally getting closely to multiple target distributions. By contrast, while MOO-SVGD also uses the MOO [7] to update the particles , their employed repulsive term encourages the particle diversity without any theoretical-guaranteed principle to control the repulsive term, hence it can force the particles to scatter on the multiple distributions. In fact, they aim to profile the whole Pareto front, which is preferred when users want to obtain a collection of diverse Pareto optimal solutions with different trade-offs among all tasks.

Furthermore, it expects that our MT-SGD globally moves the set of particles to the *joint high-likelihood region* for all target distributions. Therefore, we do not claim our MT-SGD as a method to diversify the solution on a Pareto front for user preferences, as in [15, 17]. Alternatively, our MT-SGD can generate diverse particles on the so-called *Pareto common* (i.e., the joint high-likelihood region for all target distributions). We argue and empirically demonstrate that by finding and diversifying

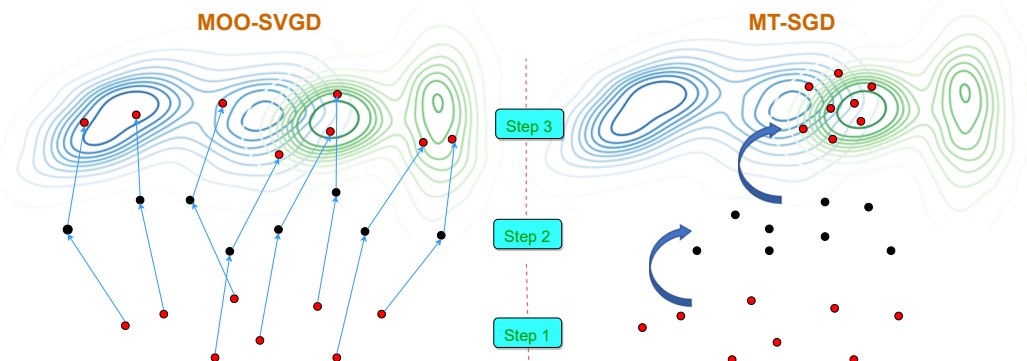

Figure 2: Our MT-SGD moves the particles from one distribution to another distribution to globally get closer to two target distributions (i.e., the blue and green ones). Differently, MOO-SVGD uses MOO [7] to move the particles individually and independently. The diversity is enforced by the repulsive forces among particles. There is no principle to control these repulsive forces, hence they can push the particles scattering on two distributions.

the particles on Pareto common for the multiple posterior inferences, our MT-SGD can outperform the baselines on Bayesian-inference metrics such as the ensemble accuracy and the calibration error.

Moreover, MOO-SVGD is *not computationally efficient* when the number of particles is high because it requires solving an independent quadratic programming problem for each particle (cf. Section 4.1.1 and Figure 3 for the experiment on a synthetic dataset). Instead, our work solves a single quadratic programming problem for all particles, then update them accordingly. We verify this computational improvement and present the training time of all baselines in the supplement material.

## 3 Application to Multi-Task Learning

For multi-task learning, we assume to have $K$ tasks $\{\mathcal{T}_i\}_{i=1}^K$ and a training set $\mathbb{D} = \{(x_i, y_{i1}, ..., y_{iK})\}_{i=1}^N$, where $x_i$ is a data example and $y_{i1}, ..., y_{iK}$ are the labels for the tasks. The model for each task $\theta^j = [\alpha, \beta^j], j = 1, ..., K$ consists of the *shared part* $\alpha$ and *non-shared part* $\beta^j$ targeting the task $j$. The posterior $p(\theta^j \mid \mathbb{D})$ for each task reads

$$p(\theta^j \mid \mathbb{D}) \propto p(\mathbb{D} \mid \theta^j) p(\theta^j) \propto \prod_{i=1}^N p(y_{ij} \mid x_i, \theta^j)$$

$$\propto \prod_{i=1}^N \exp\{-\ell(y_{ij}, x_i; \theta^j)\} = \exp\left\{-\sum_{i=1}^N \ell(y_{ij}, x_i; \theta^j)\right\},$$

where $\ell$ is a loss function and the predictive likelihood $p(y_{ij} \mid x_i, \theta^j) \propto \exp\{-\ell(y_{ij}, x_i; \theta^j)\}$ is examined. Note that the prior $p(\theta^j)$ here is retained from previous studies [13, 15], which is a uniform and non-informative prior and can be treated as a constant term in our formulation.

For our approach, we maintain a set of models $\theta_m = [\theta_m^j]_{j=1}^K$ with $m = 1, ..., M$, where $\theta_m^j = [\alpha_m, \beta_m^j]$. At each iteration, given the non-shared parts $[\beta^j]_{j=1}^K$ with $\beta^j = [\beta_m^j]_{m=1}^M$, we sample the shared parts from the multiple distributions $p(\alpha \mid \beta^j, \mathbb{D}), j = 1, ..., K$ as

$$\alpha_m \sim p(\alpha \mid \beta^j, \mathbb{D}) \propto p(\alpha, \beta^j \mid \mathbb{D}) \propto p(\theta^j \mid \mathbb{D}) \propto \exp\left\{-\sum_{i=1}^N \ell(y_{ij}, x_i; \theta^j)\right\}. \quad (6)$$

We now apply our proposed MT-SGD to sample the shared parts $[\alpha_m]_{m=1}^M$ from the multiple distributions defined in (6) as

$$\alpha_m = \alpha_m + \epsilon \sum_{j=1}^K w_j^* \phi_j^*(\alpha_m), \quad (7)$$

where $\phi_j^* (\alpha_m) = \frac{1}{M} \sum_{t=1}^{M} \left[ k_1 (\alpha_t, \alpha_m) \nabla_{\alpha_t} \log p \left( \alpha_t \mid \beta_t^j, \mathbb{D} \right) + \nabla_{\alpha_t} k_1 (\alpha_t, \alpha_m) \right]$ and $w^* = [w_k^*]_{k=1}^{K}$ are the weights received from solving the quadratic programming problem. Here we note that $\nabla_{\alpha_t} \log p \left( \alpha_t \mid \beta_t^j, \mathbb{D} \right)$ can be estimated via the batch gradient of the loss using Equation (6).

Given the updated shared parts $[\alpha_m]_{m=1}^{M}$, for each task $j$, we update the corresponding non-shared parts $\left[ \beta_m^j \right]_{m=1}^{M}$ by sampling

$$\beta_m^j \sim p \left( \beta^j \mid \alpha, \mathbb{D} \right) \propto p \left( \beta^j, \alpha \mid \mathbb{D} \right) \propto p \left( \theta^j \mid \mathbb{D} \right) \propto \exp \left\{ - \sum_{i=1}^{N} \ell \left( y_{ij}, x_i; \theta^j \right) \right\}. \qquad (8)$$

We now apply SVGD [14] to sample the non-shared parts $\left[ \beta_m^j \right]_{m=1}^{M}$ for each task $j$ from the distribution defined in (8) as

$$\beta_m^j = \beta_m^j + \epsilon \psi_j^* \left( \beta_m^j \right), \qquad (9)$$

where $\psi_j^* \left( \beta_m^j \right) = \frac{1}{M} \sum_{a=1}^{M} \left[ k_2 \left( \beta_a^j, \beta_m^j \right) \nabla_{\beta_a^j} \log p \left( \beta_a^j \mid \alpha_a, \mathbb{D} \right) + \nabla_{\beta_a^j} k_2 \left( \beta_a^j, \beta_m^j \right) \right]$ with which the term $\nabla_{\beta_a^j} \log p \left( \beta_a^j \mid \alpha_a, \mathbb{D} \right)$ can be estimated via the batch loss gradient using Equation (8).

---

**Algorithm 2** Pseudocode for multi-task learning MT-SGD.

---

**Input:** A training set $\mathbb{D} = \{ (x_i, y_{i1}, ..., y_{iK}) \}_{i=1}^{N}$.
**Output:** The models $\theta_m = \left[ \theta_m^j \right]_{j=1}^{K}$ with $m = 1, ..., M$, where $\theta_m^j = \left[ \alpha_m, \beta_m^j \right]$.
  1: Initialize a set of particles $\theta_{1:M} \sim q_0$ .
  2: **for** $epoch = 1$ to $\#epoch$ **do**
  3:   **for** $iter = 1$ to $\#iter$ **do**
  4:     Update the shared parts $[\alpha_m]_{m=1}^{M}$ using Equation. (7).
  5:     **for** $j = 1$ to $K$ **do**
  6:       Update the non-shared part $\left[ \beta_m^j \right]_{m=1}^{M}$ using Equation. (9).
  7:     **end for**
  8:   **end for**
  9: **end for**
 10: **return** $\theta_{1:M}$.

---

Algorithm 2 summarizes the key steps of our multi-task MT-SGD. Basically, we alternatively update the shared parts given the non-shared ones and vice versa.

## 4 Experiments

In this section, we verify our MT-SGD by evaluating its performance on both synthetic and real-world datasets. For our experiments, we use the RBF kernel $k (\theta, \theta') = \exp \left\{ - \| \theta - \theta' \|_2^2 / (2\sigma^2) \right\}$. The detailed training and configuration are given in the supplementary material. Our codes are available at https://github.com/VietHoang1512/MT-SGD.

### 4.1 Experiments on Toy Datasets

#### 4.1.1 Sampling from Multiple Distributions

We first qualitatively analyze the behavior of the proposed method on sampling from three target distributions. Each target distribution is a mixture of two Gaussians as $p_i (\theta) = \pi_{i1} \mathcal{N} (\theta \mid \mu_{i1}, \Sigma_{i1}) + \pi_{i2} \mathcal{N} (\theta \mid \mu_{i2}, \Sigma_{i2})$ $(i = 1, 2, 3)$ where the mixing proportions $\pi_{i1} = 0.7, \forall i$, $\pi_{i2} = 0.3, \forall i$, the means $\mu_{11} = [4, -4]^T, \mu_{12} = [0, 0.5]^T, \mu_{21} = [-4, 4]^T, \mu_{22} = [0.5, 0]^T$, and $\mu_{31} = [-3, -3]^T, \mu_{32} = [0, 0]^T$, and the common covariance matrix $\Sigma_{ij} = \begin{bmatrix} 0.5 & 0 \\ 0 & 0.5 \end{bmatrix}, i = 1, 2, 3$ and $j = 1, 2$. It can be seen from Figure 3 that there is a common high-density region spreading

around the origin. The fifty particles are drawn randomly in the space, and the initialization is retained across experiments for a fair comparison.

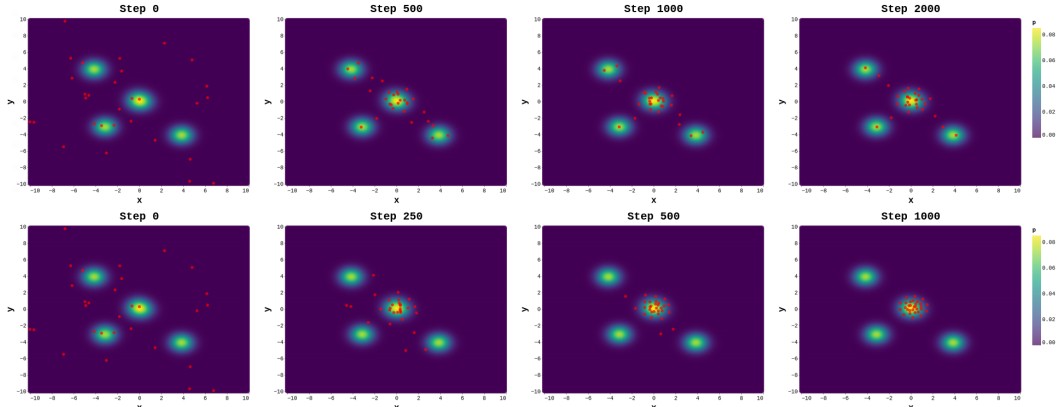

Figure 3: Sampling from three mixtures of two Gaussian distributions with a joint high-likelihood region. We run MOO-SVGD (top) and MT-SGD (bottom) to update the initialized particles (left-most figures) until convergence using Adam optimizer [11]. While MOO-SVGD transports the initialized particles scattering on the distributions, MT-SGD perfectly drives them to diversify in the region of interest.

Figure 3 shows the updated particles by MOO-SVGD and MT-SGD at selected iterations, we observe that the particles from MOO-SVGD spread out and tend to characterize all the modes, some of them even scattered along trajectories due to the conflict in optimizing multiple objectives. By contrast, our method is able to find and cover the common high density region among target distributions with well-distributed particles, which illustrates the basic principles of MT-SGD. Additionally, at the $1,000$-th step, the training time for ours is 0.23 min, whereas that for MOO-SVGD is 1.63 min. The reason is that MOO-SVGD requires solving an independent quadratic programming problem for each particle at each step.

### 4.1.2 Multi-objective Optimization

The previous experiment illustrates that MT-SGD can be used to sample from multiple target distributions, we next test our method on the other low-dimensional multi-objectives OP from [29]. In particular, we use the two objectives ZDT3, whose Pareto front consists of non-contiguous convex parts, to show our method simultaneously minimizes both objective functions. Graphically, the simulation results from Figure 4 show the difference in the convergence behaviors between MOO-SVGD and MT-SGD: the solution set achieved by MOO-SVGD covers the entire Pareto front, while ours distributes and diversifies on the three middle curves (mostly concentrated in the middle curve) which are the Pareto common having low values for two objective functions in ZDT3.

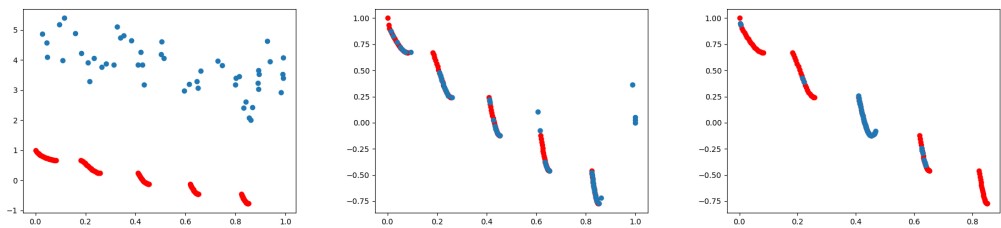

Figure 4: Solutions obtained by MOO-SVGD (mid) and MT-SGD (right) on ZDT3 problem after 10,000 steps, with blue points representing particles and red curves indicating the Pareto front. As expected, from initialized particles (left), MOO-SVGD's solution set widely distributes on the whole Pareto front while the one of MT-SGD concentrates around middle curves (mostly the middle one).

## 4.2 Experiments on Real Datasets

### 4.2.1 Experiments on Multi-Fashion+Multi-MNIST Datasets

We apply the proposed MT-SGD method on multi-task learning, following Algorithm 2. Our method is validated on different benchmark datasets: (i) Multi-Fashion+MNIST [23], (ii) Multi-MNIST, and (iii) Multi-Fashion. Each of them consists of 120,000 training and 20,000 testing images generated from MNIST [12] and FashionMNIST [27] by overlaying an image on top of another: one in the top-left corner and one in the bottom-right corner. Lenet [12] (22,350 params) is employed as the backbone architecture and trained for 100 epochs with SGD in this experimental setup.

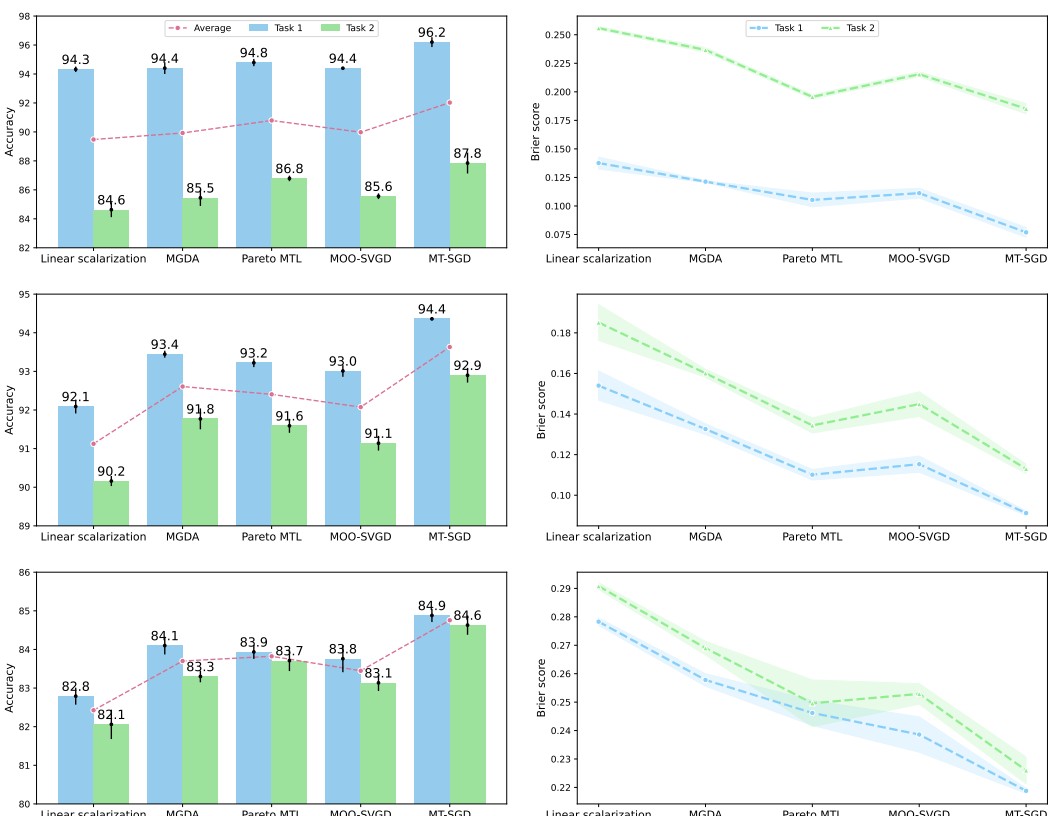

Figure 5: Results on Multi-Fashion+MNIST (left), Multi-MNIST (mid), and Multi-Fashion (right). We report the ensemble accuracy (*higher is better*) and the Brier score (*lower is better*) over 3 independent runs, as well as the standard deviation (the error bars and shaded areas in the figures).

**Baselines:** In multi-task experiments, the introduced MT-SGD is compared with state-of-the-art baselines including MGDA [24], Pareto MTL [13], MOO-SVGD [15]. We note that to reproduce results for these baselines, we either use the author's official implementation released on GitHub or ask the authors for their codes. For MOO-SVGD and Pareto MTL, the reported result is from the ensemble prediction of five particle models. Additionally, for linear scalarization and MGDA, we train five particle models independently with different initializations and then ensemble these models.

**Evaluation metrics:** We compare MT-SGD against baselines regarding both average accuracy and predictive uncertainty. Besides the commonly used accuracy metric, we measure the quality and diversity of the particle models by relying on two other popular Bayesian metrics: Brier score [3, 20] and expected calibration error (ECE) [6, 19].

From Figure 5, we observe that MT-SGD consistently improves model performance across all tasks in both accuracy and Brier score by large margins, compared to existing techniques in the literature. The network trained using linear scalarization, as expected, produces inferior ensemble results while utilizing MOO techniques helps yield better performances. Overall, our proposed method surpasses the second-best baseline by at least 1% accuracy in any experiment. Furthermore, Table 1 provides a

comparison between these methods in terms of expected calibration error, in which MT-SGD also consistently provides the lowest expected calibration error, illustrating our method's ability to obtain well-calibrated models (the accuracy is closely approximated by the produced confidence score). It is also worth noting that while Pareto MTL has higher accuracy, MOO-SVGD produces slightly better calibration estimation.

Table 1: Expected calibration error (%) (num_bin = 10) on Multi-MNIST, Multi-Fashion and Multi-Fashion+MNIST datasets over 3 runs. We use the **bold** font to highlight the best results

| Dataset | Task | Linear scalarization | MGDA | Pareto MTL | MOO-SVGD | MT-SGD |
|---|---|---|---|---|---|---|
| Multi-Fashion+MNIST | Top left | $21.33 \pm 0.83$ | $19.91 \pm 0.26$ | $9.44 \pm 0.65$ | $9.47 \pm 0.89$ | $\mathbf{4.65 \pm 0.11}$ |
| | Bottom right | $17.76 \pm 0.60$ | $16.29 \pm 1.35$ | $4.73 \pm 0.46$ | $4.95 \pm 0.49$ | $\mathbf{3.17 \pm 0.20}$ |
| Multi-MNIST | Top left | $17.37 \pm 0.62$ | $15.29 \pm 0.49$ | $5.45 \pm 0.85$ | $5.37 \pm 0.51$ | $\mathbf{3.28 \pm 0.20}$ |
| | Bottom right | $18.09 \pm 1.11$ | $16.87 \pm 0.67$ | $7.34 \pm 1.08$ | $6.74 \pm 0.50$ | $\mathbf{4.00 \pm 0.19}$ |
| Multi-Fashion | Top left | $15.86 \pm 1.20$ | $14.48 \pm 0.95$ | $8.55 \pm 0.69$ | $5.48 \pm 0.53$ | $\mathbf{3.80 \pm 0.38}$ |
| | Bottom right | $15.98 \pm 1.32$ | $14.70 \pm 1.63$ | $9.01 \pm 1.77$ | $6.11 \pm 0.54$ | $\mathbf{4.47 \pm 0.21}$ |

### 4.2.2 Experiment on CelebA Dataset

In this experiment, we verify the significance of MT-SGD on a larger neural network: Resnet18 [10], which consists of 11.4M parameters. We take the first 10 binary classification tasks and randomly select a subset of 40k images from the CelebA dataset [16]. Note that in this experiment, we consider Single task, in which 10 models are trained separately and serves as a strong baseline.

Table 2: Results on CelebA dataset, regarding accuracy and expected calibration error. For the full names of the tasks, please refer to our supplementary material. While MGDA trains a single model only to adapt on all tasks, reported performance of MOO-SVGD and MT-SGD is the ensemble results from five particle models.

| | Method | 5S | AE | Att | BUE | Bald | Bangs | BL | BN | BlaH | BloH | Average |
|---|---|---|---|---|---|---|---|---|---|---|---|---|
| Acc (%) | Single task | 91.8 | 84.6 | **80.3** | 81.9 | 98.8 | 94.8 | 85.8 | 81.3 | 89.6 | 94.2 | 88.3 |
| | MGDA | 91.8 | 84.0 | 79.0 | 81.3 | 98.6 | 94.6 | 83.6 | 81.6 | 89.8 | 93.8 | 87.8 |
| | MOO-SVGD | 92.3 | 84.2 | 78.9 | 81.2 | 98.9 | 94.5 | **86.4** | 80.0 | 90.8 | 94.8 | 88.2 |
| | MT-SGD | **92.6** | **84.8** | **80.3** | **82.9** | **99.1** | **95.2** | 86.3 | **82.6** | **91.1** | **95.0** | **89.0** |
| ECE (%) | Single task | 3.3 | 2.4 | 4.4 | 3.9 | 0.7 | 1.6 | 5.7 | 6.5 | 3.1 | 1.1 | 3.3 |
| | MGDA | 1.4 | 1.1 | 3.5 | 7.3 | **0.3** | 1.8 | 6.9 | 5.4 | 2.1 | 1.2 | 3.1 |
| | MOO-SVGD | 2.8 | 1.9 | 3.1 | 5.6 | **0.3** | **0.5** | **4.7** | 3.3 | **1.3** | 1.3 | 2.5 |
| | MT-SGD | **1.2** | **1.4** | **1.7** | **2.3** | 0.6 | 1.7 | 6.8 | **1.2** | 2.1 | **0.9** | **2.0** |

The performance comparison of the mentioned models in CelebA experiment is shown in Table 2. As clearly seen from the upper part of the table, MT-SGD performs best in all tasks, except in BL, where MOO-SVGD is slightly better (86.4% vs 86.3%). Moreover, our method matches or beats Single task - the second-best baseline in all tasks. Regarding the well-calibrated uncertainty estimates, ensemble learning methods exhibit better results. In particular, MT-SGD and MOO-SVGD provide the best calibration performances, which are 2% and 2.5%, respectively, which emphasizes the importance of efficient ensemble learning for enhanced calibration.

## 5 Conclusion

In this paper, we propose Stochastic Multiple Target **S**ampling Gradient Descent (MT-SGD), allowing us to sample the particles from the joint high-likelihood of multiple target distributions. Our MT-SGD is theoretically guaranteed to simultaneously reduce the divergences to the target distributions. Interestingly, the asymptotic analysis of our MT-SGD reduces exactly to the multi-objective optimization. We conduct comprehensive experiments to demonstrate that by driving the particles to the Pareto common (the joint high-likelihood of multiple target distributions), our MT-SGD can outperform the baselines on the ensemble accuracy and the well-known Bayesian metrics such as the expected calibration error and the Brier score.

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
