# Supplementary Material For
# Stochastic Multiple Target Sampling Gradient Descent

These appendices provide supplementary details and results of MT-SGD, including our theory development and additional experiments. This consists of the following sections:

- Appendix 1 contains the proofs and derivations of our theory development.
- Appendix 2 contains the network architectures, experiment settings of our experiments and additional ablation studies.

## 1 Proofs of Our Theory Development

### 1.1 Derivations for the Taylor expansion formulation

We have

$$\nabla_\epsilon D_{KL}\left(q^{[T]}\|p_i\right)\Big|_{\epsilon=0} = -\left\langle\phi,\psi_i\right\rangle_{\mathcal{H}_k^d}. \tag{1}$$

*Proof of Equation (1):* Since $T$ is assumed to be an invertible mapping, we have the following equations:

$$D_{KL}\left(q^{[T]}\|p_i\right) = D_{KL}\left(T\#q\|p_i\right) = D_{KL}(q\|T^{-1}\#p_i)$$

and

$$D_{KL}(q\|T^{-1}\#p_i) = D_{KL}(q\|T^{-1}\#p_i)\big|_{\epsilon=0} + \epsilon\nabla_\epsilon D_{KL}(q\|T^{-1}\#p_i)\big|_{\epsilon=0} + O(\epsilon^2). \tag{2}$$

According to the change of variables formula, we have $T^{-1}\#p_i(\theta) = p_i(T(\theta))|\det\nabla_\theta T(\theta)|$, then:

$$D_{KL}(q\|T^{-1}\#p_i) = \mathbb{E}_{\theta\sim q}[\log q(\theta) - \log p_i(T(\theta)) - \log|\det\nabla_\theta T(\theta)|].$$

Using this, the first term in Equation (2) is rewritten as:

$$\begin{aligned}
D_{KL}(q\|p_i) &= D_{KL}(T\#q\|p_i)\big|_{\epsilon=0} = D_{KL}(q\|T^{-1}\#p_i)\big|_{\epsilon=0}\\
&= \mathbb{E}_{\theta\sim q}[\log q(\theta) - \log p_i(\theta) - \log|\det\nabla_\theta\theta|] = \mathbb{E}_{\theta\sim q}[\log q(\theta) - \log p_i(\theta)]. \tag{3}
\end{aligned}$$

Similarly, the second term in Equation (2) could be expressed as:

$$\begin{aligned}
\nabla_\epsilon D_{KL}(q\|T^{-1}\#p_k)\big|_{\epsilon=0} &= \mathbb{E}_{\theta\sim q}\left[\nabla_\epsilon\log q(\theta) - \nabla_\epsilon\log p_i(T(\theta)) - \nabla_\epsilon\log|\det\nabla_\theta T(\theta)|\right]\Big|_{\epsilon=0}\\
&= -\mathbb{E}_{\theta\sim q}\left[\nabla_\epsilon\log p_i(T(\theta)) + \nabla_\epsilon\log|\det\nabla_\theta T(\theta)|\right]\Big|_{\epsilon=0}\\
&= -\mathbb{E}_{\theta\sim q}\left[\nabla_T\log p_i(T(\theta))\nabla_\epsilon T(\theta)\right]\Big|_{\epsilon=0}\\
&\quad -\mathbb{E}_{\theta\sim q}\left[\frac{1}{|\det\nabla_\theta T(\theta)|}\frac{|\det\nabla_\theta T(\theta)|}{\det\nabla_\theta T(\theta)}\nabla_\epsilon\det\nabla_\theta T(\theta)\right]\Big|_{\epsilon=0}\\
&\qquad\qquad\qquad\qquad\qquad\qquad\qquad\qquad\qquad\qquad\qquad\qquad \tag{4}\\
&= -\mathbb{E}_{\theta\sim q}\left[\nabla_T\log p_i(T(\theta))\phi(\theta)\right]\big|_{\epsilon=0}\\
&\quad -\mathbb{E}_{\theta\sim q}p\left[\frac{\det\nabla_\theta T(\theta)\operatorname{tr}((\nabla_\theta T(\theta)^{-1}\nabla_\epsilon\nabla_\theta T(\theta))}{\det\nabla_\theta T(\theta\theta)}\right]\Big|_{\epsilon=0}\\
&= -\mathbb{E}_{\theta\sim q}[\nabla_\theta\log p_i(\theta)\phi(\theta) + \operatorname{tr}(\nabla_\theta\phi(\theta))].
\end{aligned}$$

It could be shown from the reproducing property of the RKHS that $\phi_i(\theta) = \langle \phi_i(\cdot), k(\theta, \cdot) \rangle_{\mathcal{H}_k}$, then we find that

$$\frac{\partial \phi_i(\theta)}{\partial \hat{\theta}_i} = \left\langle \phi_i(\cdot), \frac{\partial k(\theta, \cdot)}{\partial \hat{\theta}_i} \right\rangle_{\mathcal{H}_k}. \tag{5}$$

Let $U_{d \times d} = \nabla_\theta \phi(\theta)$ whose $u_i^T$ denotes the $i^{th}$ row vector and the particle $\theta \in \mathbb{R}^d$ is represented by $\{\hat{\theta}\}_{i=1}^d$, the row vector $u_i^T$ is given by:

$$u_i^T := \frac{\partial \phi_i(\theta)}{\partial \theta} = \frac{\partial \phi_i(\theta)}{\partial(\hat{\theta}_1, \hat{\theta}_2, \ldots, \hat{\theta}_d)} = \left[ \frac{\partial \phi_i(\theta)}{\partial \hat{\theta}_1} ; \frac{\partial \phi_i(\theta)}{\partial \hat{\theta}_2} ; \ldots ; \frac{\partial \phi_i(\theta)}{\partial \hat{\theta}_d} \right]. \tag{6}$$

Combining Property (5) and Equation (6), we have:

$$u_i^T := \left[ \frac{\partial \phi_i(\theta)}{\partial \hat{\theta}_1} ; \frac{\partial \phi_i(\theta)}{\partial \hat{\theta}_2} ; \ldots ; \frac{\partial \phi_i(\theta)}{\partial \hat{\theta}_d} \right]$$

$$= \left[ \left\langle \phi_i(\cdot), \frac{\partial k(\theta, \cdot)}{\partial \hat{\theta}_1} \right\rangle_{\mathcal{H}_k} ; \left\langle \phi_i(\cdot), \frac{\partial k(\theta, \cdot)}{\partial \hat{\theta}_2} \right\rangle_{\mathcal{H}_k} ; \ldots ; \left\langle \phi_i(\cdot), \frac{\partial k(\theta, \cdot)}{\partial \hat{\theta}_d} \right\rangle_{\mathcal{H}_k} \right]. \tag{7}$$

Substituting Equation (7) to Equation (4), the linear term of the Taylor expansion could be derived as:

$$\nabla_\epsilon D_{KL}(q \| T^{-1} \# p_i) \big|_{\epsilon=0} = -\mathbb{E}_{\theta \sim q}[\nabla_\theta \log p_i(\theta) \phi(\theta) + \text{tr}(\nabla_\theta \phi(\theta))]$$

$$= -\mathbb{E}_{\theta \sim q}\left[ \sum_{j=1}^d \langle \phi_j(\cdot), k(\theta, \cdot) \rangle_{\mathcal{H}_k} (\nabla_\theta \log p_i(\theta))_j + \frac{\partial \phi_j(\theta)}{\partial \hat{\theta}_j} \right)\right]$$

$$= -\sum_{j=1}^d \mathbb{E}_{\theta \sim q}\left[ \langle \phi_j(\cdot), k(\theta, \cdot)(\nabla_\theta \log p_i(\theta))_j \rangle_{\mathcal{H}_k} + \left\langle \phi_j(\cdot), \left(\frac{\partial k(\theta, \cdot)}{\partial \theta}\right)_j \right\rangle_{\mathcal{H}_k} \right]$$

$$= -\sum_{j=1}^d \left\langle \phi_j(\cdot), \mathbb{E}_{\theta \sim q}\left[ k(\theta, \cdot)(\nabla_\theta \log p_i(\theta))_j + \left(\frac{\partial k(\theta, \cdot)}{\partial \theta}\right)_j \right] \right\rangle_{\mathcal{H}_k}$$

$$= - \langle \phi(\cdot), \psi(\cdot) \rangle_{\mathcal{H}_k^d},$$

where $(v)_j$ denotes the $j$-th element of $v$ and $\psi(\cdot) \in \mathcal{H}_k^d$ is a matrix whose $j^{th}$ column vector is given by

$$\mathbb{E}_{\theta \sim q}\left[ k(\theta, \cdot)(\nabla_\theta \log p_i(\theta))_j + \left(\frac{\partial k(\theta, \cdot)}{\partial \theta}\right)_j \right].$$

In other word, the formula of $\psi(\cdot)$ becomes

$$\mathbb{E}_{\theta \sim q}\left[ k(\theta, \cdot) \nabla_\theta \log p_i(\theta) + \frac{\partial k(\theta, \cdot)}{\partial \theta} \right].$$

As a consequence, we obtain the conclusion of Equation (1).

## 1.2  Proof of Lemma 1

Before proving this lemma, let us re-state it:

**Lemma 1.** *Let $w^*$ be the optimal solution of the optimization problem $w^* = \operatorname*{argmin}_{w \in \Delta_K} w^T U w$ and $\phi^* = \sum_{i=1}^K w_i^* \phi_i^*$, where $\Delta_K = \left\{ \pi \in \mathbb{R}_+^K : \|\pi\|_1 = 1 \right\}$ and $U \in \mathbb{R}^{K \times K}$ with $U_{ij} = \langle \phi_i^*, \phi_j^* \rangle_{\mathcal{H}_k^d}$, then we have*

$$\langle \phi^*, \phi_i^* \rangle_{\mathcal{H}_k^d} \geq \|\phi^*\|_{\mathcal{H}_k^d}^2, i = 1, \ldots, K.$$

*Proof.* For arbitrary $\epsilon \in [0, 1]$ and $u \in \Delta_K$, then $\omega := \epsilon u + (1 - \epsilon) w^* \in \Delta_K$, we thus have the following inequality:

$$w^{*T} U w^* \leq \omega^T U \omega$$

$$= (\epsilon u + (1 - \epsilon) w^*)^T U (\epsilon u + (1 - \epsilon) w^*)$$

$$= (w^* + \epsilon(u - w^*))^T U (w^* + \epsilon(u - w^*))$$

$$= w^{*T} U w^* + 2\epsilon w^{*T} U (u - w^*) + \epsilon^2 (u - w^*)^T U (u - w^*),$$

which is equivalent to

$$0 \leq 2\epsilon w^{*T} U(u - w^*) + \epsilon^2 (u - w^*)^T U(u - w^*). \tag{8}$$

Hence $w^{*T} U(u - w^*) \geq 0$, since otherwise the R.H.S of inequality (8) will be negative with sufficiently small $\epsilon$. By that, we arrive at

$$w^{*T} U w^* \leq w^{*T} U u.$$

By choosing $u$ to be a one hot vector at $i$, we obtain the conclusion of Lemma 1.

## 1.3 Derivations for the matrix $U_{ij}$'s formulation in Equation (3)

We have

$$\phi_i^* (\cdot) = \mathbb{E}_{\theta \sim q} \left[ k(\theta, \cdot) \nabla \log p_i(\theta) + \nabla k(\theta, \cdot) \right],$$
$$\phi_j^* (\cdot) = \mathbb{E}_{\theta' \sim q} \left[ k(\theta', \cdot) \nabla \log p_j(\theta') + \nabla k(\theta', \cdot) \right].$$

Therefore, we find that

$$U_{ij} = \langle \phi_i^*, \phi_j^* \rangle_{\mathcal{H}_k^d} = \mathbb{E}_{\theta, \theta' \sim q} \left[ \langle k(\theta, \cdot), k(\theta', \cdot) \rangle_{\mathcal{H}_k} \sum_{l=1}^{d} \nabla_{\theta_l} \log p_i(\theta) \nabla_{\theta_l'} \log p_j(\theta') \right.$$

$$+ \sum_{l=1}^{d} \nabla_{\theta_l} \log p_i(\theta) \left\langle k(\theta, \cdot), \nabla_{\theta_l'} k(\theta', \cdot) \right\rangle_{\mathcal{H}_k} + \sum_{l=1}^{d} \nabla_{\theta_l'} \log p_j(\theta') \langle k(\theta', \cdot), \nabla_{\theta_l} k(\theta, \cdot) \rangle_{\mathcal{H}_k}$$

$$\left. + \sum_{l=1}^{d} \left\langle \nabla_{\theta_l} k(\theta, \cdot), \nabla_{\theta_l'} k(\theta', \cdot) \right\rangle_{\mathcal{H}_k} \right],$$

which is equivalent to

$$U_{ij} = \mathbb{E}_{\theta, \theta' \sim q} \left[ k(\theta, \theta') \langle \nabla \log p_i(\theta), \nabla \log p_j(\theta') \rangle \right.$$

$$+ \left\langle \nabla \log p_i(\theta), \frac{\partial k(\theta, \theta')}{\partial \theta'} \right\rangle + \left\langle \nabla \log p_j(\theta'), \frac{\partial k(\theta, \theta')}{\partial \theta} \right\rangle +$$

$$\left. + \sum_{l=1}^{d} \left\langle \nabla_{\theta_l} k(\theta, \cdot), \nabla_{\theta_l'} k(\theta', \cdot) \right\rangle_{\mathcal{H}_k} \right].$$

Now, note that

$$\langle k(\theta, .), \varphi(.) \rangle_{\mathcal{H}_k} = \varphi(\theta),$$

hence we gain

$$\langle \nabla_{\theta_l} k(\theta, .), \varphi(.) \rangle_{\mathcal{H}_k} = \nabla_{\theta_l} \varphi(\theta),$$

which follows that

$$\left\langle \nabla_{\theta_l} k(\theta, \cdot), \nabla_{\theta_l'} k(\theta', \cdot) \right\rangle_{\mathcal{H}_k} = \nabla_{\theta_l, \theta_l'}^2 k(\theta, \theta'),$$

$$\sum_{l=1}^{d} \left\langle \nabla_{\theta_l} k(\theta, \cdot), \nabla_{\theta_l'} k(\theta', \cdot) \right\rangle_{\mathcal{H}_k} = \sum_{l=1}^{d} \nabla_{\theta_l, \theta_l'}^2 k(\theta, \theta') = \text{tr} \left( \frac{\partial^2 k(\theta, \theta')}{\partial \theta \partial \theta'} \right).$$

Putting these results together, we obtain that

$$U_{ij} = \langle \phi_i^*, \phi_j^* \rangle_{\mathcal{H}_k^d} = \mathbb{E}_{\theta, \theta' \sim q} \left[ k(\theta, \theta') \langle \nabla \log p_i(\theta), \nabla \log p_j(\theta') \rangle \right.$$

$$\left. + \left\langle \nabla \log p_i(\theta), \frac{\partial k(\theta, \theta')}{\partial \theta'} \right\rangle + \left\langle \nabla \log p_j(\theta'), \frac{\partial k(\theta, \theta')}{\partial \theta} \right\rangle + \text{tr} \left( \frac{\partial^2 k(\theta, \theta')}{\partial \theta \partial \theta'} \right) \right].$$

As a consequence, we obtain the conclusion of Equation (3).

## 1.4 Proof of Theorem 2

Before proving this theorem, let us re-state it:

**Theorem 2.** $w \in \Delta_K$ such that $\sum_{i=1}^{K} w_i \phi_i^* = 0$, given a sufficiently small step size $\epsilon$, all KL divergences w.r.t. the target distributions are strictly decreased by at least $A \|\phi^*\|_{\mathcal{H}_k^d}^2 > 0$ where $A$ is a positive constant.

*Proof.* We have for all $i = 1, ..., K$ that

$$D_{KL}\left(q^{[T]}\|p_i\right) = D_{KL}\left(q\|p_i\right) + \nabla_\epsilon D_{KL}\left(q^{[T]}\|p_i\right)\Big|_{\epsilon=0} \epsilon + O_i\left(\epsilon^2\right)$$

$$= D_{KL}\left(q\|p_i\right) - \langle \phi, \psi_i \rangle_{\mathcal{H}_k^d} \epsilon + O_i\left(\epsilon^2\right)$$

$$\leq D_{KL}\left(q\|p_i\right) - \|\phi^*\|_{\mathcal{H}_k^d}^2 \epsilon + O_i\left(\epsilon^2\right).$$

Because $\lim_{\epsilon \to 0} \frac{O_i(\epsilon^2)}{\epsilon^2} = B_i$, there exists $\alpha_i > 0$ such that $|\epsilon| < \alpha_i$ implies $\left|O_i(\epsilon^2)\right| < \frac{3}{2}|B_i|\epsilon^2$. By choosing, $B = \frac{3}{2}\max_i |B_i|$ and $\alpha = \min_i \alpha_i$, we arrive at for all $\epsilon < \alpha$ and all $i$

$$D_{KL}\left(q^{[T]}\|p_i\right) < D_{KL}\left(q\|p_i\right) - \|\phi^*\|_{\mathcal{H}_k^d}^2 \epsilon + B\epsilon^2.$$

Finally, by choosing sufficiently small $\epsilon > 0$, we reach the conclusion of the theorem. □

# 2 Implementation Details

In this appendix, we provide implementation details regarding the empirical evaluation in the main paper along with additional comparison experiments.

## 2.1 Experiments on Toy Datasets

### 2.1.1 Sampling from Multiple Distribution

In this experiment, the three target distributions are created as presented in the main paper. The particle's coordinates are randomly sampled from the normal distribution $\mathcal{N}(0, 5)$. Adam optimizer [5] with learning rate of $3e-2$ and $\beta_1 = 0.9, \beta_2 = 0.999$ is used to update the particles. MOO-SVGD and MT-SGD converged after 2000 and 1000 iterations, respectively.

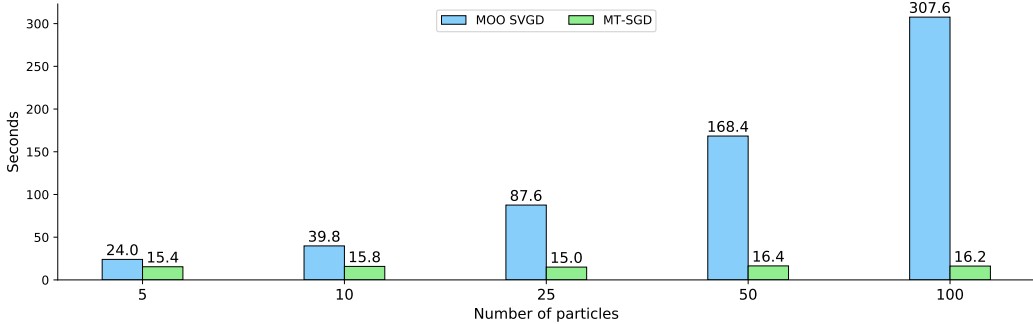

Figure 1: Running time of MT-SGD and MOO-SVGD for 1000 steps on: Intel(R) Xeon(R) CPU @ 2.20GHz CPU and Tesla T4 16GB VRAM GPU. Results are averaged over 5 runs.

We also measure the running time between MOO-SVGD and our proposed method when varying the number of particles from 5 to 100. In Figure 1, we plot the time consumption when running MOO-SVGD and MT-SGD in 1000 iterations. As can be seen that, MOO-SVGD runtime grows linearly with the number of particles, since it requires solving separate quadratic problems (Algorithm 1) for each particle. By contrast, there is only one quadratic programming problem solving in our proposed method, which significantly reduces time complexity, especially when the number of particles is high.

### 2.1.2 Multi-objective Optimization

ZDT-3 [13] is a classic benchmark problem in multi-objective optimization with 30 variables $\theta = (\theta_1, \theta_2, \ldots, \theta_{30})$ with a number of disconnected Pareto-optimal fronts. This problem is given by:

$$\min f_1(\theta),$$
$$\min f_2(\theta) = g(\theta)h(f_1(\theta), g(\theta)),$$

where

$$f_1(\theta) = \theta_1,$$
$$g(\theta) = 1 + \frac{9}{29}\sum_{i=2}^{30}\theta_i,$$
$$h(f_1, g) = 1 - \sqrt{\frac{f_1}{g}} - \frac{f_1}{g}\sin(10\pi f_1),$$
$$0 \le \theta_i \le 1, i = 1, 2, \ldots, 30.$$

The Pareto optimal solutions are given by

$$0 \le \theta_1 \le 0.0830,$$
$$0.1822 \le \theta_1 \le 0.257,$$
$$0.4093 \le \theta_1 \le 0.4538,$$
$$0.6183 \le \theta_1 \le 0.6525,$$
$$0.8233 \le \theta_1 \le 0.8518,$$
$$\theta_i = 0 \text{ for } i = 2, \ldots, 30.$$

For ZDT3 experiment, we utilize Adam [5], learning rate $5e-4$ and update the 50 particles for 10000 iterations as in the comparative baseline [8].

### 2.1.3 Multivariate regression

We consider the SARCOS regression dataset [12], which contains 44,484 training samples and 4,449 testing samples with 21 input variables and 7 outputs (tasks). The train-test split in [10] is kept, with 40,036 training examples, 4,448 validation examples, and 4,449 test examples. We replicate the neural network architecture from [10] as follows: $21 \times 256\,\text{FC} \to \text{ReLU} \to 256 \times 256\,\text{FC} \to \text{ReLU} \to 256 \times 256\,\text{FC} \to \text{ReLU} \to 256 \times 7\,\text{FC}$ (139,015 params). The network is optimized by Adam [5] optimizer for 1000 epochs, with $\beta_1 = 0.9, \beta_2 = 0.999$, and the learning rate of $1e-4$. All experimental results are obtained by running five times with different seeds.

Table 1: Mean square errors of MT-SGD and competing methods on SARCOS dataset [12]. We take the best checkpoint in each approach based on the validation score. Results are averaged over 5 runs, and we highlight the best method for each task in **bold**.

|  | Method | Task 1 | Task 2 | Task 3 | Task 4 | Task 5 | Task 6 | Task 7 | Average |
|---|---|---|---|---|---|---|---|---|---|
| | MGDA | 0.025 | 0.2789 | 0.0169 | 0.0026 | 1.158 | 0.264 | 0.005 | 0.25 |
| Validation | MOO-SVGD | 0.0177 | 0.2182 | 0.0113 | 0.0013 | 1.241 | 0.2292 | 0.0025 | 0.2459 |
| | MT-SGD | **0.0173** | **0.2124** | **0.0112** | **0.0012** | **1.110** | **0.2208** | **0.0024** | **0.2251** |
| | MGDA | 0.0082 | 0.0675 | 0.0038 | 0.0009 | 0.2635 | 0.0455 | 0.0018 | 0.0559 |
| Test | MOO-SVGD | 0.0043 | 0.0586 | 0.0019 | 0.0003 | 0.2584 | 0.0365 | 0.0007 | 0.0515 |
| | MT-SGD | **0.0037** | **0.0515** | **0.0018** | **0.0002** | **0.2097** | **0.0318** | **0.0005** | **0.0428** |

Regarding the baselines for this experiment, we compare our method against MGDA [11], MOO-SVGD [8]. We empirically set the batch size as 512 and $M = 5$ particles in MT-SGD and competing methods. The mean square error for each task and the average results are shown in Table 1. We find that our method achieves the lowest error on all tasks, with the largest gap on Task 4. MT-SGD outperforms the second-best method, MOO-SVGD, with 0.2251 vs. 0.2459 on validation set and 0.0428 vs. 0.0515 on test set.

## 2.2 Experiments on Real Datasets

### 2.2.1 Experiments on Multi-Fashion+Multi-MNIST Datasets

We follow the same training protocol with previous work [7, 8, 11], Lenet [6] is trained in 100 epochs with SGD optimizer. The input images are in size $36 \times 36$ and the training batch size is 256.

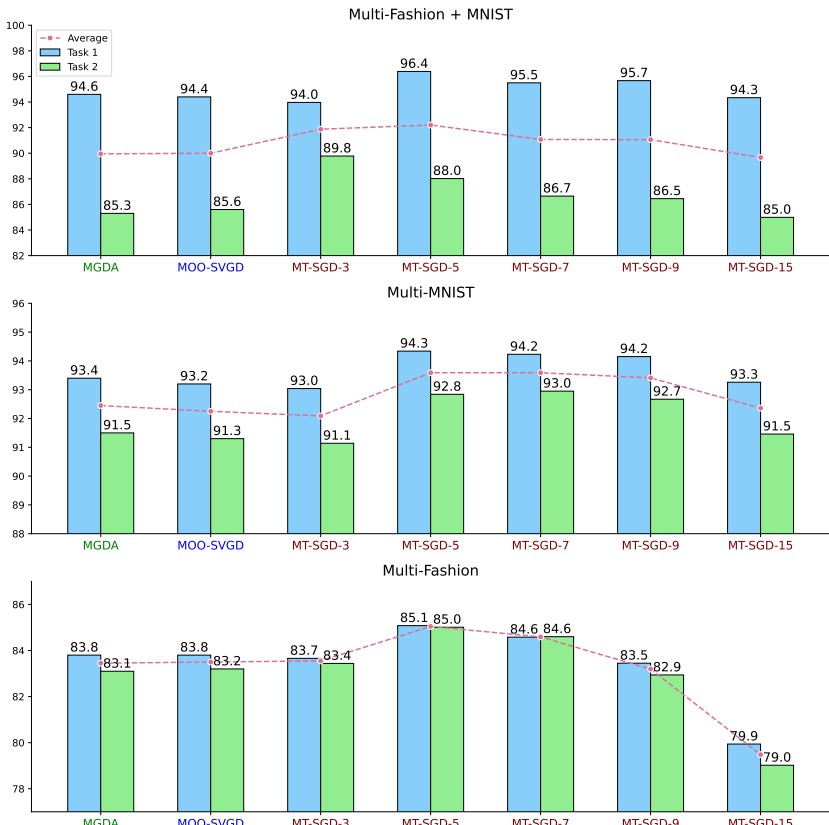

Figure 2: Average accuracy (%) when varying the number of particles from 3 to 15. MT-SGD-m denotes our method using m particle networks.

We now study the performance of our proposed method against variation in the number of particles by conducting more experiments on Multi-MNIST/Fashion/Fashion+MNIST datasets. We vary the number of neural networks in $3, 5, 7, 9, 15$ and present the accuracy scores in Figure 2. From the results, a simple conclusion that can be derived is that increasing the number of particle networks from $3 \rightarrow 5$ improves the performance in all three datasets, surpassing the other two baselines, while further increasing this hyperparameter does not help.

**Metrics:** In the main paper, we compare our proposed method against baselines in terms of Brier score and expected calibration error (ECE). We here provide more details on how to calculate these metrics. Assumed that the training dataset $\mathcal{D}$ consists of $N$ i.i.d examples $\mathcal{D} = \{x_n, y_n\}_{n=1}^N$ where $y_n \in \{1, 2, \ldots, K\}$ denotes corresponding labels for K-class classification problem. Let $p(y = c | x_i)$ be the predicted confidence that $x_i$ belongs to class C.

- **Brier score:** The Brier score is computed as the squared error between a predicted probability $p(y|x_i)$ and the one-hot vector ground truth:

$$BS = \frac{1}{N} \sum_{i=1}^N \sum_{c=1}^K \left( \mathbf{1}_{y_i = c} - p(y = c | x_i) \right)^2$$

- **Expected calibration error:** Partitioning predictions into $M$ equally-spaced bins $B_m = \left( \frac{m-1}{M}, \frac{m}{M} \right] (m = 1, 2 \ldots, M)$, the expected calibration error is computed as the average

gap between the accuracy and the predicted confidence within each bin:

$$\text{ECE} = \sum_{m=1}^{M} \frac{|B_m|}{N} |\text{acc}(B_m) - \text{conf}(B_m)|$$

where $\text{acc}(B_m)$ and $\text{conf}(B_m)$ denote the accuracy and confidence of bin $B_m$

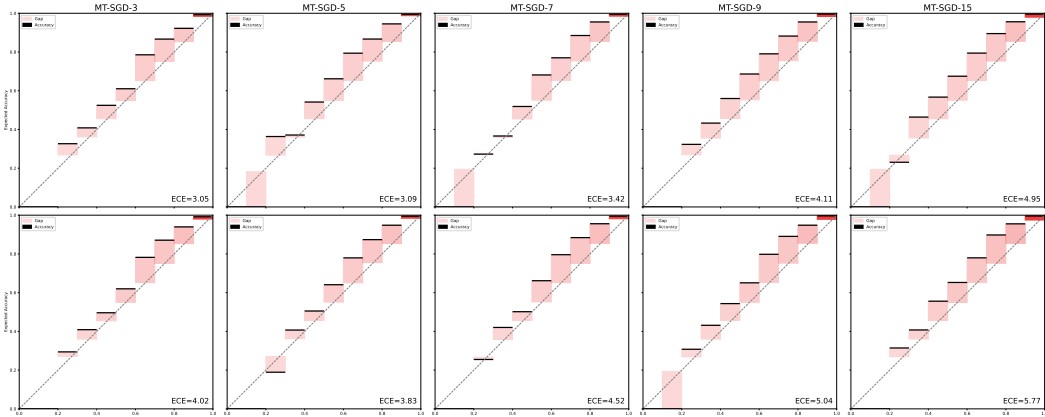

Figure 3: Expected Calibration Error (%) when varying the number of particles from 3 to 15 on Multi-MNIST. MT-SGD-m denotes our method using m particle networks. We set the number of bins equal to 10 throughout the experiments.

Figure 3 displays ECE as a function of the number of particles. Similar to the average accuracy metric, the Expected Calibration Error reduces when we increase the number of particle networks from 3 to 5 yet does not decrease in the cases of MT-SGD-7, MT-SGD-9 and MT-SGD-15.

**Computational complexity of MT-SGD:** From the complexity point of view, MT-SGD introduces a marginal computational overhead compared to MGDA since it requires calculating the matrix $U$, which has a complexity $O(K^2 M^2 d)$, where the number of particles $M$ is usually set to a small positive integer. However, on the one hand, computing $U$'s entries can be accelerated in practice by calculating them in parallel since there is no interaction between them during forward pass. On the other hand, the computation of the back-propagation is typically more costly than the forward pass. Thus, the main bottlenecks in our method lie on the backward pass and solving the quadratic programming problem - which is an iterative method [3, 11].

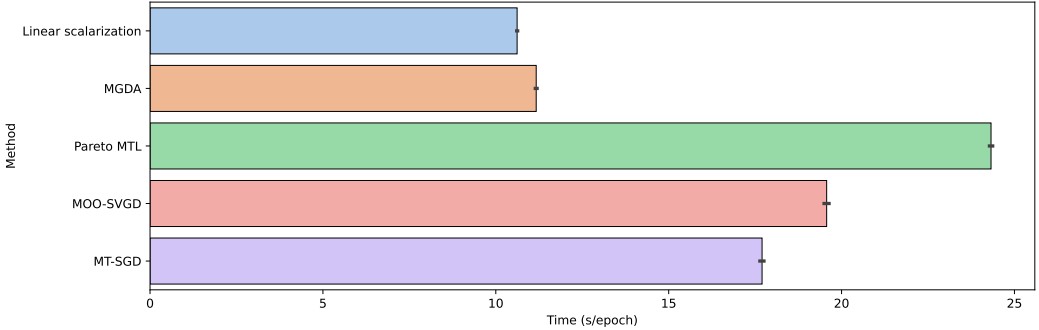

Figure 4: Running time on each epoch of MT-SGD and compared baselines on Multi-MNIST dataset. Results are averaged over 5 runs, with the standard deviation reported by error bars.

As a final remark in the Multi-Fashion+Multi-MNIST experiment, we compare our methods against baselines in terms of the required running time in a single epoch and plot the result in Figure 4. We observe that in our experiments on Multi-MNIST dataset, the computation time of methods that enforce the diversity of obtained models is higher than that of the methods that do not (Pareto MTL,

MOO-SVGD, MT-SGD vs Linear scalarization, MGDA). Nevertheless, this is a small price to pay for the major gain in the ensemble performance, since our training involves the interaction between models to impose diversity. Compared to Pareto MTL and MOO-SVGD, the running time of our proposed MT-SGD is considerably better ( 2s less than MOO-SVGD and  6s less than Pareto MTL).

### 2.2.2   Experiment on CelebA Dataset

We performed our experiments on the CelebA dataset [9], which contains images annotated with 40 binary attributes. Resnet-18 backbone [2] without the final layer as a shared encoder and a 2048 x 2 dimensional fully connected layer for each task is employed as in [11]. We train this network for 100 epochs with Adam optimizer [5] of learning rate $5e - 4$ and batch size $64$. All images are resized to $64 \times 64 \times 3$.

Due to space constraints, we report only the abbreviation of each task in the main paper, their full names are presented below.

Table 2: CelebA binary classification tasks full names.

| 5S | AE | Att | BUE | Bald | Bangs | BL | BN | BlaH | BloH |
|---|---|---|---|---|---|---|---|---|---|
| 5 O'clock Shadow | Arched Eyebrows | Attractive | Bags Under Eyes | Bald | Bangs | Big Lips | Big Nose | Black Hair | Blond Hair |

Now we investigate the effectiveness of our proposed MT-SGD method on the whole CelebA dataset, compared with prior work: Uniform scaling: minimizing the uniformly weighted sum of objective functions, Single task: train separate models individually for each task, Uncertainty [4]: adaptive reweighting with balanced uncertainty, Gradnorm [1]: balance the loss functions via gradient magnitude and MGDA [11]. The results from previous work are reported in [1]. For MGDA, we use their officially released codebase at `https://github.com/isl-org/MultiObjectiveOptimization`. For a fair comparison, we run the code with five different random seeds and present the obtained scores in Figure 5.

Following [11], we divide 40 target binary attributes into two subgroups: hard and easy tasks for easier visualization. As can be seen from Figure 5, we observe that the naively trained Uniform scaling has relatively low performance on many tasks, e.g. "Mustache", "Big Lips", "Oval Face". Compared to other baselines, our proposed method significantly reduces the prediction error in almost all the tasks, especially on "Goatee", "Double Chi" and "No Beard". The detailed result for each target attribute can be found in Table 3.

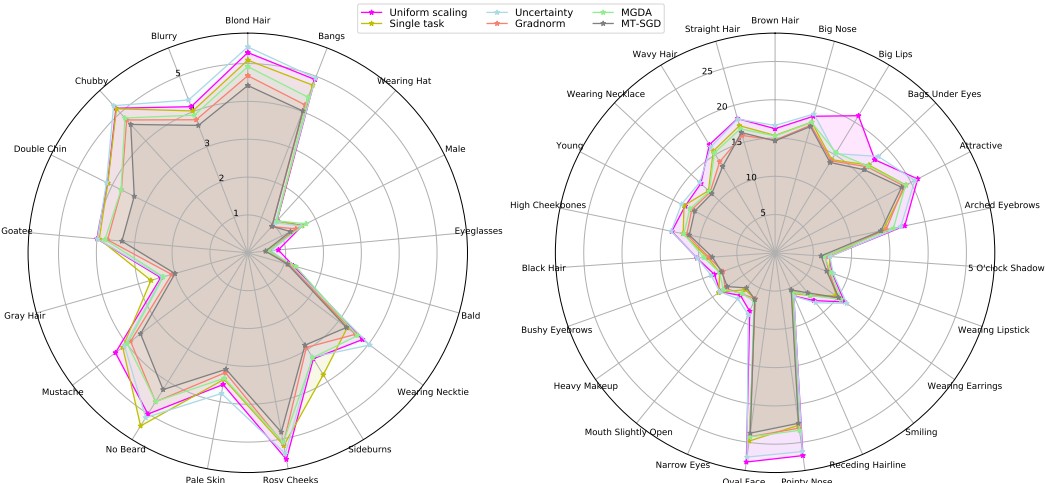

Figure 5: Radar charts of prediction error on CelebA [9] for each individual binary classification task. We divide attributes into two sets: easy tasks on the left, difficult tasks on the right, as in [11].

Table 3: Average performance (lower is better) of each target attribute for all baselines. We use the **bold** font to highlight the best-obtained score in each task.

| Attribute | Uniform scaling | Single task | Uncertainty | Gradnorm | MGDA | MT-SGD |
|---|---|---|---|---|---|---|
| 5 O'clock Shadow | 7.11 | 7.16 | 7.18 | 6.54 | 6.47 | **6.03** |
| Arched Eyebrows | 17.30 | 14.38 | 16.77 | 14.80 | 15.80 | **14.11** |
| Attractive | 20.99 | 19.25 | 20.56 | 18.97 | 19.21 | **18.62** |
| Bags Under Eyes | 17.82 | 16.79 | 18.45 | 16.47 | 16.60 | **15.91** |
| Bald | 1.25 | 1.20 | 1.17 | 1.13 | 1.32 | **1.09** |
| Bangs | 4.91 | 4.75 | 4.95 | 4.19 | 4.41 | **4.02** |
| Big Lips | 20.97 | 14.24 | 15.17 | 14.07 | 15.32 | **13.82** |
| Big Nose | 18.53 | 17.74 | 18.84 | 17.33 | 17.70 | **17.14** |
| Black Hair | 10.22 | 8.87 | 10.19 | 8.67 | 9.31 | **8.22** |
| Blond Hair | 5.29 | 5.09 | 5.44 | 4.68 | 4.92 | **4.42** |
| Blurry | 4.14 | 4.02 | 4.33 | 3.77 | 3.90 | **3.61** |
| Brown Hair | 16.22 | 15.34 | 16.64 | 14.73 | 15.27 | **14.63** |
| Bushy Eyebrows | 8.42 | 7.68 | 8.85 | **7.23** | 7.69 | 7.42 |
| Chubby | 5.17 | 5.15 | 5.26 | 4.75 | 4.82 | **4.59** |
| Double Chin | 4.14 | 4.13 | 4.17 | 3.73 | 3.74 | **3.35** |
| Eyeglasses | 0.81 | 0.52 | 0.62 | 0.56 | 0.54 | **0.47** |
| Goatee | 4.00 | 3.94 | 3.99 | 3.72 | 3.79 | **3.34** |
| Gray Hair | 2.39 | 2.66 | 2.35 | 2.09 | 2.32 | **2.00** |
| Heavy Makeup | 8.79 | 9.01 | 8.84 | 8.00 | 8.29 | **7.65** |
| High Cheekbones | 13.78 | 12.27 | 13.86 | 11.79 | 12.18 | **11.45** |
| Male | 1.61 | 1.61 | 1.58 | 1.42 | 1.72 | **1.26** |
| Mouth Slightly Open | 7.18 | 6.20 | 7.73 | 6.91 | 6.86 | **5.91** |
| Mustache | 4.38 | 4.14 | 4.08 | 3.88 | 3.99 | **3.55** |
| Narrow Eyes | 8.32 | 6.57 | 8.80 | **6.54** | 6.88 | 6.64 |
| No Beard | 5.01 | 5.38 | 5.12 | 4.63 | 4.62 | **4.25** |
| Oval Face | 27.59 | 24.82 | 26.94 | 24.26 | 24.28 | **23.78** |
| Pale Skin | 3.54 | 3.40 | 3.78 | 3.22 | 3.37 | **3.13** |
| Pointy Nose | 26.74 | 22.74 | 26.21 | 23.12 | 23.41 | **22.48** |
| Receding Hairline | 6.14 | 5.82 | 6.17 | 5.43 | 5.52 | **5.28** |
| Rosy Cheeks | 5.55 | 5.18 | 5.40 | 5.13 | 5.10 | **4.82** |
| Sideburns | 3.29 | 3.79 | 3.24 | 2.94 | 3.26 | **2.87** |
| Smiling | 8.05 | 7.18 | 8.40 | 7.21 | 7.19 | **6.74** |
| Straight Hair | 18.21 | 17.25 | 18.15 | **15.93** | 16.82 | 16.32 |
| Wavy Hair | 16.53 | 15.55 | 16.19 | 13.93 | 15.28 | **13.19** |
| Wearing Earrings | 11.12 | **9.76** | 11.46 | 10.17 | 10.57 | 10.17 |
| Wearing Hat | 1.15 | 1.13 | 1.08 | **0.94** | 1.14 | 0.95 |
| Wearing Lipstick | 7.91 | 7.56 | 8.06 | 7.47 | 7.76 | **7.15** |
| Wearing Necklace | 13.27 | 11.90 | 13.47 | 11.61 | 11.75 | **11.32** |
| Wearing Necktie | 3.80 | 3.29 | 4.04 | 3.57 | 3.63 | **3.27** |
| Young | 13.25 | 13.40 | 13.78 | 12.26 | 12.53 | **11.83** |
| Average | 9.62 | 8.77 | 9.53 | 8.44 | 8.73 | **8.17** |