# OpenReview forum: "Stochastic Multiple Target Sampling Gradient Descent"
_NeurIPS.cc/2022/Conference — NeurIPS 2022 Accept_

### Official Review · Reviewer_6tPS · 2022-07-10

**Rating:** 7
**Confidence:** 2
**Soundness:** 3 good
**Presentation:** 3 good
**Contribution:** 3 good

**Summary:**

In this work, the authors introduce Stochastic Multiple Target Sampling Gradient Descent (MT-SGD), an extension of Stein variational gradient descent (SVGD) that tries to approximately sample from multiple distributions *simultaneously*. The authors then demonstrate that MT-SGD is well suited for Bayesian multi-task learning.

**Questions:**

1. In section 2.2.1 in the appendix, there is an interesting phenomenon in Figure 2 where increasing the number of particles, $M$, seems to **deteriorate** the performance substantially.  A prior, I don't see why this should be the case. I would appreciate it if the authors could clear this up.
2. In  section on Multi-task learning, the authors should state that they are using a non-informative prior for the parameters of the model.

**Limitations:**

While on the checklist the authors stated that they talk about the limitations in the appendix, I don't see it anywhere. To me, the two biggest limitations are the scaling with the number of tasks, $K$, and the number of particles, $M$ (which is inherent to SVGD). I think the authors should mention it in the work, though it doesn't detract from the novelty of the work.

**Strengths And Weaknesses:**

# Strengths

MT-SGD is a natural extension of SVGD that is theoretically motivated. Moreover, the authors were also able to demonstrate for RBF kernels, as the scale goes to infinity, MT-SGD reduces to multiple objectivation optimization which is amazing! Moreover, the computational complexity of the algorithm seems to be on the order of SVGD, with the main difference being solving the quadratic programming problem for finding the optimal combination of the gradients. While I'm not familiar with the SVGD and multi-task learning literature, this approach seems both highly novel and theoretically grounded. The experiments section is also compelling as well.

# Weaknesses

Firstly, I'm a little confused about the Multi-task learning formulation. Given the data, one would want to compute the following posterior distribution
$$ p(\alpha, \beta^1, \ldots, \beta^K \mid \mathbb{D})  \propto p(\alpha) \prod_{j=1}^K p(\beta^j) \prod_{i=1}^N p(y_{ij} \mid x_i, \alpha, \beta^j) $$
Conditioned on the non-shared parts, $\beta^1, \ldots, \beta^K$, the posterior for the shared parts is
$$ p(\alpha \mid \mathbb{D}, \beta^1, \ldots, \beta^J) \propto p(\alpha) \prod_{j=1}^K \prod_{i=1}^N p(y_{ij} \mid x_i, \alpha, \beta^j)  $$
where we can see that this conditional posterior already attempts to find an $\alpha$ that is good for all tasks simulatensouly. Moreover, just regular SVGD can be used for this. I don't see the advantage, nor the intuition, of formulating the problem as $K$ individual posteriors. I think it would strengthen the approach if they authors compared SVGD on the posterior I have outlined.

Next, while the experiments section is great, it is **very** concerning that there are no error bars for all the experiments in section 4.2, especially given how close the accuracies are for each of these tasks. Thus, I think it is imperative the authors run the experiments over a number of realizations to make me more confident in their results.

Please let me know if I have misunderstood anything! I will also be happy to raise my score if the authors get back to me about my concerns.

---

> ### Author Response · Authors · 2022-08-02
> **Authors' Response to Reviewer 6tPS (1/3)**
>
> We thank the reviewer for constructive feedbacks on our paper. Below we provide further clarification with respect to your main concerns:
>
> **Multi-task learning formulation**
>
> Thank you for your comment, it is true that using your proposed derivation is a common way to tackle the multi-task learning problem:
>
> $$p\left(\alpha \mid \mathbb{D}, \beta^{1}, \ldots, \beta^{J}\right) \propto p(\alpha) \prod_{j=1}^{K} \prod_{i=1}^{N} p\left(y_{i j} \mid x_{i}, \alpha, \beta^{j}\right)$$
> $$ \Leftrightarrow \log p\left(\alpha \mid \mathbb{D}, \beta^{1}, \ldots, \beta^{J}\right) \propto \log p(\alpha) + \sum_{j=1}^{K} p\left(y_{j} \mid x, \alpha, \beta^{j}\right)$$
>
> Therefore, it can be clearly seen that this formulation is equivalent to uniformly weighting all tasks, which is a straightforward and commonly-used approach in practice.
>
> However, our motivation in this work is to incorporate the MOO scheme into the sampling problem, allowing us to sample the shared part simultaneously in the Pareto common of multiple tasks. By contrast, simply using uniform weights for all tasks could lead to a deterioration in performance, as carefully investigated in previous literature: (MGDA, ParetoMTL, MOO-SVGD, and etc.).

---

> > ### Author Response · Authors · 2022-08-02
> > **Authors' Response to Reviewer 6tPS (2/3)**
> >
> > Furthermore, to extend our current experimental evaluation and include the comparison against your suggested uniform scaling method, we have run the experiment on the full large-scale CelebA dataset and reported the details in the supplementary material, with the performance of previous work reported from MGDA paper:
> >
> >
> > | Attribute           | Uniform scaling | Single task | Uncertainty | Gradnorm | MGDA    | MT-SGD (our)  |
> > |-|-|-|-|-|-|-|
> > | 5 O'clock Shadow    |  7.11           |  7.16       |  7.18       |  6.54    |  6.47   |  **6.03**   |
> > | Arched Eyebrows     |  17.30          |  14.38      |  16.77      |  14.80   |  15.80  |  **14.11**  |
> > | Attractive          |  20.99          |  19.25      |  20.56      |  18.97   |  19.21  |  **18.62** |
> > | Bags Under Eyes     |  17.82          |  16.79      |  18.45      |  16.47   |  16.60  |  **15.91** |
> > | Bald                |  1.25           |  1.20       |  1.17       |  1.13    |  1.32   | **1.09**  |
> > | Bangs               |  4.91           |  4.75       |  4.95       |  4.19    |  4.41   |  **4.02**  |
> > | Big Lips            |  20.97          |  14.24      |  15.17      |  14.07   |  15.32  |  **13.82**  |
> > | Big Nose            |  18.53          |  17.74      |  18.84      |  17.33   |  17.70  |  **17.14**  |
> > | Black Hair          |  10.22          |  8.87       |  10.19      |  8.67    |  9.31   | **8.22**   |
> > | Blond Hair          |  5.29           |  5.09       |  5.44       |  4.68    |  4.92   | **4.42**  |
> > | Blurry              |  4.14           |  4.02       |  4.33       |  3.77    |  3.90   |  **3.61**   |
> > | Brown Hair          |  16.22          |  15.34      |  16.64      |  14.73   |  15.27  |  **14.63**  |
> > | Bushy Eyebrows      |  8.42           |  7.68       |  8.85       |  **7.23**    |  7.69   |  7.42   |
> > | Chubby              |  5.17           |  5.15       |  5.26       |  4.75    |  4.82   |  **4.59**  |
> > | Double Chin         |  4.14           |  4.13       |  4.17       |  3.73    |  3.74   |  **3.35**   |
> > | Eyeglasses          |  0.81           |  0.52       |  0.62       |  0.56    |  0.54   |  **0.47**   |
> > | Goatee              |  4.00           |  3.94       |  3.99       |  3.72    |  3.79   |  **3.34**   |
> > | Cray Hair           |  2.39           |  2.66       |  2.35       |  2.09    |  2.32   |  **2.00**   |
> > | Heavy Makeup        |  8.79           |  9.01       |  8.84       |  8.00    |  8.29   | **7.65**   |
> > | High Cheekbones     |  13.78          |  12.27      |  13.86      |  11.79   |  12.18  |  **11.45** |
> > | Male                |  1.61           |  1.61       |  1.58       |  1.42    |  1.72   | **1.26**   |
> > | Mouth Slightly Open |  7.18           |  6.20       |  7.73       |  6.91    |  6.86   |  **5.91**   |
> > | Mustache            |  4.38           |  4.14       |  4.08       |  3.88    |  3.99   |  **3.55**   |
> > | Narrow Eves         |  8.32           |  6.57       |  8.80       |  **6.54**   |  6.88   |  6.64   |
> > | No Beard            |  5.01           |  5.38       |  5.12       |  4.63    |  4.62   |  **4.25**  |
> > | Oval Paw            |  27.59          |  24.82      |  26.94      |  24.26   |  24.28  | **23.78**  |
> > | Pale Skin           |  3.54           |  3.40       |  3.78       |  3.22    |  3.37   |  **3.13**   |
> > | Pointy Nose         |  26.74          |  22.74      |  26.21      |  23.12   |  23.41  |  **22.48** |
> > | Receding Hairline   |  6.14           |  5.82       |  6.17       |  5.43    |  5.52   | **5.28**   |
> > | Rosy Cheeks         |  5.55           |  5.M        |  5.40       |  5.13    |  5.10   |  **4.82**   |
> > | Sideburns           |  3.29           |  3.79       |  3.24       |  2.94    |  3.26   |  **2.87**   |
> > | Smiling             |  8.05           |  7.M        |  8.40       |  7.21    |  7.19   |  **6.74**  |
> > | Straight Hair       |  18.21          |  17.25      |  18.15      |  **15.93**   |  16.82  | 16.32|
> > | Wavy Hair           |  16.53          |  15.55      |  16.19      |  13.93   |  15.28  |  **13.19**  |
> > | Wearing Earrings    |  11.12          |  **9.76**       |  11.46      |  10.17   |  10.57  | 10.17  |
> > | Wearing Hat         |  1.15           |  1.13       |  1.08       |  **0.94**    |  1.14   |  0.95   |
> > | Wearing Lipstick    |  7.91           |  7.56       |  8.06       |  7.47    |  7.76   |  **7.15**  |
> > | Wearing Necklace    |  13.27          |  11.90      |  13.47      |  11.61   |  11.75  |  **11.32**  |
> > | Wearing Necktie     |  3.80           |  3.29       |  4.04       |  3.57    |  3.63   |  **3.27**  |
> > | Young               |  13.25          |  13.40      |  13.78      |  12.26   |  12.53  |  **11.83**  |
> > | Average             |  9.62           |  8.77       |  9.53       |  8.44    |  8.73   |  **8.17**  |

---

> > > ### Author Response · Authors · 2022-08-02
> > > **Authors' Response to Reviewer 6tPS (3/3)**
> > >
> > > **Error bar**
> > >
> > > We already plotted the accuracy scores in the upper part of Figure 5. Regarding your suggestion of running the experiments over a number of realizations, we have run all the methods on three different random seeds and updated the result on the revision. Thanks for pointing this out to make our comparison more reliable.
> > >
> > >
> > > **Increasing the number of particles leads to performance drop**
> > >
> > > An intuitive explanation for this behaviour is that the more particles are included in training, the more among them are ejected from the high-likelihood region by the repulsive force. Therefore, these models can hurt the ensemble performance. Also, we provide some empirical examples with 3/10/50 particles (Figures 1-3) here: https://sites.google.com/view/mt-sgd-rebuttal/. Thus, one can observe from Figure 3 that many of the fifty particles are outside the common mode region.
> > >
> > > **Prior of model parameter**
> > >
> > > Following the previous work, the initial weights distributions $p(θ)$ is chosen following the default initialization of torch (e.g. kaiming_uniform for [fully connected](https://github.com/pytorch/pytorch/blob/master/torch/nn/modules/linear.py#L103) and [conv layers](https://github.com/pytorch/pytorch/blob/master/torch/nn/modules/conv.py#L150)). Thank you for your suggestion, we have clarified this in the revision.
> > >
> > > **Complexity**
> > >
> > > As can be seen from Algorithm 1 and Equation from Line 142, the main difference in terms of complexity between our proposed MT-SGD with the MGDA baseline is that it requires computing the $U$ matrix, which has a $O(K^2M^2d)$ complexity where the number of particles $M$ is usually set to a small positive integer. Furthermore, computing $U$'s entries can be accelerated in practice by calculating them in parallel, as there is no interaction between them during forward pass.
> > >
> > > However, the computation of the back-propagation is typically more costly than the forward pass. Thus, the main bottlenecks in our method lie on the backward pass and solving the quadratic programming problem - which requires an iterative method. We have run one more experiment to measure the running time in each epoch of MT-SGD and baselines. The result is reported in the supplementary and presented in Figure 4: https://sites.google.com/view/mt-sgd-rebuttal/
> > >
> > > Thank you for your willingness to increase our paper's score. Please let us know if you would like us to do anything else.

---

> > > > ### Author Response · Authors · 2022-08-08
> > > > **Looking forward to hearing your further feedback**
> > > >
> > > > Dear Reviewer 6tPS,
> > > >
> > > > We would like to thank you again for spending your time evaluating our paper.
> > > >
> > > > As the discussion period is expected to conclude shortly, we look forward to hearing your feedback about whether we have addressed your concerns in the rebuttals.
> > > >
> > > > Best regards,
> > > >
> > > > Authors

---

> > > > > ### Comment · Reviewer_6tPS · 2022-08-08
> > > > > **Response**
> > > > >
> > > > > Thanks so much for the reply and I apologize for the late response! I think the response was detailed and addressed most of my concerns. I have increased my score accordingly!

---

> > > > > > ### Author Response · Authors · 2022-08-09
> > > > > > **Thank you**
> > > > > >
> > > > > > We greatly appreciate your reconsideration!
> > > > > >
> > > > > > Best regards,
> > > > > >
> > > > > > Authors.

---

### Official Review · Reviewer_yFUe · 2022-07-11

**Rating:** 6
**Confidence:** 4
**Soundness:** 3 good
**Presentation:** 3 good
**Contribution:** 3 good

**Summary:**

This paper proposed a variant of SVGD for multi-task learning. In particular, instead of driving particles towards the high-density region of a single distribution like SVGD, the proposed MT-SGD drives the particles towards the joint region of all densities. The main novelty of this paper is to propose a weighted combination of the driving directions from each SVGD such that the combined direction is guaranteed to decrease all KL divergences between the current distribution and each target distribution. Additionally, like the connection of SVGD to SGD, the author showed the connection of MT-SGD to MOO. Empirically, the proposed MT-SGD achieves better results compared to baselines consistently.

**Questions:**

**Clarity**:
1. From my understanding, the main novelty is the objective in lemma 1. Is there an intuition on why $w^TUw$ is a reasonable objective?. It seems that this objective is very similar to the one used in MOO (Eq.3). But instead replacing the derivative of $L$ with the functional derivative of $D_{KL}$. Is this true? In that case, personally, I think it would be better if MT-SGD is introduced from the MOO point of view. In this way, it is also easier for the reader to understand intuition. Also, can MT-SGD be derived based on the Pareto stationary KKT condition?

2. In Eq.1, what does the comma mean? I understand this is a direct adaptation of MOO, but can you explain this minimization, since the typical minimization is targeted at a single value, but here, we have a list of $D_{KL}$.

3. In line 49, do you mean empirical distributions?

4. In line 107, is the optimal transformation $T$ for the current step?

5. In line 112, should it be $O(\epsilon^2)$?

6. Just curious, can the objective $w^TUw$ in lemma 1 be translated to an optimization problem of the weighted combination of $D_{KL}$?

7. In line 175, I suggest using a different subscript for $w$, since $t$ is already reserved for the number of SVGD steps in algorithm 1.
8. For the captions of Figure 4, it should be red curve.

**SVGD-related**:
1. One of my concerns about using SVGD is its mode collapse problem, which has been extensively reported [1]. This means without proper tuning of the repulsive force, the particles tend to collapse to a single point in high dimensions. Have you noticed this behaviour? If not, why?
2. In line 143, the author mentioned when $\sigma \rightarrow \infty$, $U$ reduces to the inner product of the score function. But it seems that the update of MT-SGD, in this case, does not recover the MOO? Since the kernel matrix in SVGD is now a matrix with value 1, which takes the gradient of other particles into consideration.


[1]: Gong, Wenbo, Yingzhen Li, and José Miguel Hernández-Lobato. "Sliced kernelized Stein discrepancy." arXiv preprint arXiv:2006.16531 (2020).


**Limitations:**

This work does not have any potential negative societal impact. In terms of the potential limitation, the author argues that the proposed MT-SGD tries to find the Pareto common, whereas the other baselines try to find the Pareto front. I am curious under what scenario Pareto front is preferred compared to Pareto common?

**Strengths And Weaknesses:**

**Strength**:
This paper adopts the framework of MOO and extends to SVGD. The basic idea is very similar to MOO, the novelty comes from the incorporation of SVGD. The objective for searching $w^*$ can also be viewed as an adaptation of the MOO objective. In summary, the proposed approach is novel to some extent, but as mentioned by the author, this is not a completely novel framework. In terms of significance, the proposed framework demonstrates clear advantages compared to baselines and seems to be a good contribution to Bayesian multi-tasking learning.

**Weakness**:
I have checked the proofs in the appendix, which seem to be correct. Thus, my main criticism is its presentation. I will elaborate more in the question section.

---

> ### Author Response · Authors · 2022-08-02
> **Authors' Response to Reviewer yFUe**
>
> We greatly appreciate the reviewer’s detailed and constructive comments and suggestions. In the following, we provide the main response to your comments:
>
> **Clarity**
> 1. **The objective function $w^TUw$**
>
>    In our paper, the derived objective function $w^TUw$ is inspired by the MGDA (MOO) method. Specifically, MOO's gradient acts on a space of gradients, whereas our optimal pushforward function $\phi^*$ acts on $\mathcal{H}_k^d$ where $\mathcal{H}_k$ is a Reproducing Kernel Hilbert space corresponding to the kernel $k$.
>
>    In the paper, we explicitly show the connection to MOO and state our problem as a multi-objective optimization problem on a probability space, as shown in Equation (2). Additionally, Theorem 2 dictates the Pareto stationary condition and indicates that if the steep descent directions $\phi_i^*$ are linearly dependent, we reach a Pareto stationary point where the obtained pushforward function $\phi^*$ becomes the zero function and cannot help to further decrease the divergences.
>
> 2. **Comma in Equation (1)**
>
>    The comma here acts as a separator between Equation (1)  and the below notation explanation. In terms of the minimization of a list of $D_{KL}$, we are interested in simultaneously minimizing all objectives (i.e. gradually updating the particles to the common high-likelihood region). That is the reason why we cast this problem as a MOO, from which we can derive a principled method to achieve this goal. Due to space constraints, we will include more clarification in the later revision.
>
> 3. **In line 49, do you mean empirical distributions**
>
>     Yes, it is. We mean a empirical (uniform) distribution over the set of particles.
>
> 4. **In line 107, is the optimal transformation $T$ for the current step?**
>
>    Yes, it is. We explicitly present this information from Line 103 to 107 in the paper. Nevertheless, we have reminded this a second time in the revision to avoid confusion for the readers.
>
> 5. **In line 112, should it be $O(\epsilon^2)$?**
>
>    Thank you for spotting this typo, it should be $O(\epsilon^2)$. We have fixed it in the revised version.
>
> 6. **Can the objective $w^TUw$ in lemma 1 be translated to an optimization problem of the weighted combination of $D_{KL}$?**
>
>    The objective function $w^TUw$ in lemma 1 cannot be translated to an optimization problem of the weighted combination of $D_{KL}$. The reason is that although the optimal $\phi^*$ is linearly dependent on the steepest descent direction $\phi^*$, when we apply them to pushforward the current distribution, the linear dependency cannot be preserved to the KL divergences.
>
> 7. **In line 175, using a different subscript for $w$**
>
>    Thanks for your suggestion. We have updated the revised version accordingly.
>
> 8. **The captions of Figure 4**
>
>    Thanks. We have corrected it in the revised version.
>
> **SVGD-related**
>
> 1. **Mode collapse problem**
>
>    Thanks for your question. We do not confront the mode collapsing problem on the synthetic datasets and multi-task learning. We conjecture that in our approach, the particles need to sufficiently spread out to orient multiple target distributions (or tasks) and also further avoid collapsing with the inherit repulsive forces. Therefore, the particle collapsing problem is less likely to happen. However, we certainly believe that clearly investigating the behaviour of SVGD in high-dimensional data is a necessary next step to pursue in future work.
>
> 2. **Assymtotic behavior when $\sigma$ goes to $\infty$**
>
>     Yes, in that case, the formulation might be asymptotically relevant to some extent but does not exactly reduce to MOO.
>
> **Limitation**
>
>    Finding the Pareto front is preferred when users want to obtain a collection of diverse Pareto optimal solutions with different trade-offs among all tasks. Thus, one can select their preferred trained model among the solution set at inference time. For instance, [2] utilized MOO to jointly train the main task with another auxiliary task and then selected the trained model that had an outstanding performance in the main task only. However, there might some of them are *extreme* models (i.e. works well on one task while performing poorly on the others).
>
> [2] Yim, Jonghwa, and Sang Hwan Kim. "Learning boost by exploiting the auxiliary task in multi-task domain." arXiv preprint arXiv:2008.02043 (2020).

---

> > ### Author Response · Authors · 2022-08-08
> > **Looking forward to hearing your further feedback**
> >
> > Dear Reviewer yFUe,
> >
> > We would like to thank you again for spending your time evaluating our paper.
> >
> > As the discussion period is expected to conclude shortly, we look forward to hearing your feedback about whether we have addressed your concerns in the rebuttals.
> >
> > Best regards,
> >
> > Authors

---

> > > ### Comment · Reviewer_yFUe · 2022-08-08
> > > **Reply to author's response**
> > >
> > > I appreciate the author's detailed response. It addresses most of my concerns. Personally, I think in the revised version, adapting the current presentation of the proposed method to a format like MOO will be easier for the readers to understand. For example, although the objective $w^TUw$ is inspired by MOO, it is less clear from reading the main text until the author's reply.

---

> > > > ### Author Response · Authors · 2022-08-09
> > > > **Thank you**
> > > >
> > > > We would like to thank the reviewer for spending their time evaluating our paper and providing detailed feedbacks. We will include the intuition behind our objective function in the next version of our manuscript, as you suggested.
> > > >
> > > > Best,
> > > >
> > > > Authors.

---

### Official Review · Reviewer_TqCR · 2022-07-12

**Rating:** 6
**Confidence:** 3
**Soundness:** 2 fair
**Presentation:** 3 good
**Contribution:** 3 good

**Summary:**

This paper proposes MT-SGD for multi-objective optimization problems, which leverages the idea of SVGD to generate diverse particles w.r.t. the joint high-likelihood region for all distributions. Experiments on both synthetic dataset and real dataset (Multi-Fashion+Multi-MNIST and CelebA) demonstrate the effectiveness of MT-SGD compared to MGDA, Pareto MTL and MOO-SVGD.

**Questions:**

None

**Ethics Review Area:**

["I don’t know"]

**Strengths And Weaknesses:**

Strength:
+ Writing is good and the paper is easy to follow
+ Finding diverse particles on the Pareto front is a well-motivated topic.
+ The proposed method is novel and inspiring

Weakness:
- Lack of convergence analysis of MT-SGD. Theorem 1 & 2 proves that MT-SGD reduces every KL divergence simultaneously, but it is unclear how the variational distribution $q$ or the particle set $\{\theta_i\}_{i=1}^M$ will be when it converged.
- The equation between line 166 and line 167 seems weird. It should be $p(\theta|D) \propto p(D|\theta) p(\theta)$. Besides, how the prior $p(\theta)$ is chosen in experiments?

---

> ### Author Response · Authors · 2022-08-02
> **Authors' Response to Reviewer TqCR**
>
> We greatly appreciate the reviewer’s detailed and constructive comments and suggestions. Below we address the main concerns raised in your review.
>
> 1. **Lack of convergence analysis of MT-SGD**
>
>    We totally agree with your observation. Currently, we only prove that if the particles are updated according to the proposed algorithm, at each step, KL divergences to target distributions are decreased by a certain amount. However, we do not have any theoretical results regarding the convergence analysis at this point.
>
>    Moreover, Theorem 2 indicates the Pareto stationary condition happening when our approach converges. This theorem shows that the proposed approach halts when the steepest descent directions $\phi_i^*$(s) are linearly dependent, hence the resultant pushforward function $\phi^*$ is zero. Eventually, the current distribution stays still.
>
>    Additionally, developing convergence analysis for our approach is a challenging theoretical problem for further study. At this outset, we realize it is a highly tough and challenging theoretical study. Evidently, for non-linear objective functions (e.g., a deep learning empirical loss), we need to make strong assumptions about its local smoothness to guarantee its convergence to a local minima. For non-linear multi-objective optimization, there has not been any work addressing its convergence to a Pareto stationary point, to the best of our knowledge.
>
> 2. **Equation line 166-167**
>
>    Yes, you're right. In detail, the formulation is written as $p(\theta \mid D) \propto p(D \mid \theta)$ $p(\theta)$ (following the Bayes' theorem). However, for a fair comparison, the prior (initial weights distributions) $p(θ)$ is retained from previous work (MGDA, ParetoMTL, MOO-SVGD), which is the default initialization of torch (e.g. kaiming_uniform for [fully connected](https://github.com/pytorch/pytorch/blob/master/torch/nn/modules/linear.py#L103) and [conv layers](https://github.com/pytorch/pytorch/blob/master/torch/nn/modules/conv.py#L150)). So that, we treat the prior $p(\theta)$ as a constant term and eliminate it out of the main objective for brevity.  We have updated the revised version to explicitly state this elimination.
>
> We hope that you can reconsider the review score. Please let us know if you would like us to do anything else.

---

> > ### Author Response · Authors · 2022-08-08
> > **Looking forward to hearing your further feedback**
> >
> > Dear Reviewer TqCR,
> >
> > We would like to thank you once again for spending your time evaluating our paper.
> >
> > As the discussion period is expected to conclude shortly, we look forward to hearing your feedback about whether we have addressed your concerns in the rebuttals.
> >
> > Best regards,
> >
> > Authors

---

> > > ### Comment · Reviewer_TqCR · 2022-08-09
> > > **Reply to author's response**
> > >
> > > The authors' response has addressed most of my concerns. I'll raise my score accordingly.

---

> > > > ### Author Response · Authors · 2022-08-09
> > > > **Thank you**
> > > >
> > > > Thank you for recognizing our efforts and providing constructive reviews. We greatly appreciate your endorsement.
> > > >
> > > > Best regards,
> > > >
> > > > Authors.

---

### Meta-Review · Area_Chair_kBKo · 2022-08-24

**Recommendation:** Accept
**Confidence:** Less certain

**Metareview:**

The paper presents a particle-based method to approximate multiple target distributions simultaneously. The proposed particle-updating dynamics is shown to decrease the KL to every target (makes a Pareto improvement), and the resulting particles prefer the intersection of all targets (Pareto common) which differs it from a related method (which prefers Pareto front). Although the technical framework is not completely novel (follows MGDA (MOO)), reviewers agree that the proposed method for multi-distribution approximation is inspiring and the paper has well implemented the idea.

Nevertheless, there still remains a few imprecise statements that authors need to address upon acceptance.

1. Precise meeding of Eqs. (1,2). It is impossible that a single $q$ simultaneously minimizes each individual KL in general. Reviewer yFUe mentioned this but the reply did not clearly define this. The notation/formulation needs clarification even if it follows previous work.

2. Equation below Line 114 is only true if $\phi$ is in RKHS.

3. Some statements on MOO-SVGD might be improper. It seems conflicting that MOO-SVGD “updates the particles individually and independently”, while also “employs a repulsive term”, which is an interation among particles. More precise description is expected on (MOO-SVGD) “encourages the particle diversity without any theoretical-guaranteed principle to control the repulsive term”: MOO-SVGD is not originally intended for multi-distribution approximation, and it also provides a stationary distribution characterization.


**Award:**

No

---

### Decision · Program_Chairs · 2022-09-14

Accept